# Bioadhesive and conformable bioelectronic interfaces for vasomotoricity monitoring and regulation

Xiner Wang[1,2,10], Weijian Fan[3,10], Yuxin Liu[1], Li Chen[4], Erda Zhou[1,2], Xiaoling Wei[2,5,6], Liuyang Sun[1,2,6], Bo Yu[3], Tiger H. Tao[7,8,9] ✉, Zhitao Zhou[2,5,6] ✉ & Jinyun Tan[3] ✉

The autonomic nervous system (ANS) dynamically regulates vasomotor function to maintain vascular homeostasis, yet current clinical tools lack the capacity to capture its electrophysiological basis, particularly following stent implantation. Herin, we report a bioadhesive and conformable bioelectronic (BACE) interface engineered for effective monitoring and modulation of vasomotoricity. Incorporating silk fibroin-based adhesives, the interface adheres robustly to cylindrical surfaces in aqueous conditions for up to two months. It exhibits low interfacial impedance ($6.77 \pm 2.13$ kΩ at 1 kHz) and background noises ($2.63 \pm 0.52$ μV) in vivo, enabling precise recording of varying vasomotor states. In a stent model, the interface identifies alterations in bioelectrical activity associated with vasomotor dysfunction, validated against ultrasound-derived arterial stiffness as clinical gold standards. Furthermore, we demonstrate a bidirectional system for the detection and modulation of dysfunction to restore vasomotor activity and elasticity. These findings provide insights into vasomotor electrophysiology mechanisms and underscore the therapeutic potential of bioelectronic modulation in vascular disease.

Blood vessels act as the major components of the human or animal circulatory system, performing a crucial function in maintaining the stability of blood circulation throughout the body. Among these vessels, arteries are the primary conduits through which oxygen-rich blood is delivered from the heart to peripheral tissues. The regulation of blood flow through the arteries hinges on their dynamic ability to constrict and dilate, a process that is strongly modulated by the ANS. Autonomic nerve activity is intricately linked to arterial vasomotor function, with disruptions in vascular tone frequently associated with alterations in autonomic drive. In terms of outcomes, vasomotor dysfunction can lead to hemodynamic instability, contributing to the development and progression of severe cardiovascular conditions, such as hypertension and atherosclerosis[1-4]. In response to these complications, stent implantation has emerged as a cornerstone of modern endovascular treatment, widely utilized to manage a spectrum of diseases characterized by vascular stenosis or occlusion. Millions of percutaneous coronary interventions with stent deployment are performed worldwide annually, representing the predominant

[1]2020 X-Lab, Shanghai Institute of Microsystem and Information Technology, Chinese Academy of Sciences, Shanghai, China. [2]School of Graduate Study, University of Chinese Academy of Sciences, Beijing, China. [3]Department of Vascular Surgery, Huashan Hospital of Fudan University, Shanghai, China. [4]Department of Ultrasound, Huashan Hospital of Fudan University, Shanghai, China. [5]State Key Laboratory of Transducer Technology, Shanghai Institute of Microsystem and Information Technology, Chinese Academy of Sciences, Shanghai, China. [6]School of Integrated Circuits, University of Chinese Academy of Sciences, Beijing, China. [7]Neuroxess Co. Ltd, Shanghai, China. [8]Guangdong Institute of Intelligence Science and Technology, Zhuhai, ChinaGuangdong. [9]Tianqiao and Chrissy Chen Institute for Translational Research, Shanghai, China. [10]These authors contributed equally: Xiner Wang, Weijian Fan. ✉e-mail: tiger@mail.sim.ac.cn; ztzhou@mail.sim.ac.cn; m.tan@fudan.edu.cn

revascularization procedure, and their importance continues to grow driven by an aging population and lifestyle changes[5,6]. While stent implantation is well-established for restoring vessel patency, its potential impact on vasomotor function warrants attention. A stent may contribute to complications such as thrombosis and restenosis, potentially arising from mechanical injury, impaired endothelial function, and reduced arterial wall compliance. These factors can ultimately lead to dysfunction and pathophysiological changes in vasomotor responses[7–11].

Currently, clinical evaluation of vascular status primarily relies on imaging modalities such as digital subtraction angiography (DSA)[12], Doppler ultrasound[13], computed tomography angiography[14], and magnetic resonance angiography[15]. The clinical modalities focus on vascular morphology, anatomical structure, and flow dynamics for assessing vascular lesions and patency, which fundamentally differ from electrophysiological techniques in the principles and the nature of information they provide, leaving a gap in understanding how the neural regulation of autonomic nervous system influences overall vasomotor function. Functional assessments such as pulse wave velocity and ankle-brachial index are also commonly used to detect and evaluate the severity of arterial diseases[16,17]. However, these ancillary tests are constrained by limited accuracy and resolution, rendering them insufficient for capturing the intricate dynamics of vascular function. Innovations in mechanical sensing technologies, including piezoresistive[18], capacitive[19,20], piezoelectric[21,22], and triboelectric[23] sensors, have facilitated the measurement of vascular mechanical properties such as blood pressure, flow velocity, and pulse wave velocity. However, as a process governed by autonomic nerve activity, the vasomotor function relies fundamentally on electrophysiological events, which serve as the basis for neural signal transmission and the execution of physiological responses. Recent advances in implantable bioelectronic interfaces address these limitations by establishing direct contact with target tissues, enabling more precise measurement of physiological parameters compared to traditional methods. Despite these advancements, existing technologies are insufficient for directly and reliably recording high-quality vascular electrophysiological activity, which is essential for unraveling the electrophysiological mechanisms underlying the ANS regulation of vasomotor tone.

Here, we report a bioadhesive and conformable bioelectronic interface specifically designed to adapt to the three-dimensional tubular structure of blood vessels for vasomotoricity monitoring and regulation, even within environments filled with interstitial fluid. This interface incorporates hydrophilic polyurethane (PU) into natural silk fibroin (SF) to create an adhesive and flexible substrate with high interfacial toughness and low bending stiffness. This bioadhesive substrate is positioned as the foundational supporting layer beneath the microfabricated electrodes, serving to anchor the interface to biological tissues while effectively minimizing mechanical mismatch between the implant and the dynamic vessel wall. By integrating this adhesive overcoat, the resulting bioelectronic interface adheres securely to biological tissues and conforms seamlessly to curved surfaces. This design enables intimate and stable integration with the vascular surface in fluid-rich physiological environments, supporting consistent and artifact-resistant signal acquisition. The BACE interface achieves a low interfacial impedance of $6.77 \pm 2.13\,k\Omega$ at 1 kHz and a low baseline noise of $2.63 \pm 0.52\,\mu V$ in vivo, ensuring high-fidelity recording of vascular electrophysiological (VE) activity. With its 64-channel configuration, the system offers enhanced spatial resolution, allowing for detailed tracking of vasomotor dynamics across arterial segments and accurate monitoring of vasomotor dysfunction with performance validated against the stiffness parameter, which is the clinical gold standard in arterial assessment. Furthermore, we demonstrate a bidirectional system that leverages the interface's electrical stimulation capability to detect and modulate vasomotor dysfunction, enhancing recovery from impaired vasomotor activity with corresponding improvements in arterial stiffness ($\beta$ decreased from 18.83 to 14.06). This bidirectional bioelectronic interface enables real-time detection and targeted neuromodulation of vasomotor dysfunction, which holds significant potential for elucidating the electrophysiological mechanisms underlying vasomotor function and advancing both the clinical diagnosis and therapeutic intervention of vascular diseases.

## Results

### Overview of the BACE interface

In the ANS, postganglionic fibers originating from both the sympathetic and parasympathetic ganglia extend to the arteries, exerting both direct and indirect effects on smooth muscle cells and endothelial cells. Through a series of synergistic mechanisms, these fibers modulate cellular functions and influence the activity and expression of their receptors, ultimately orchestrating vasoconstriction and vasodilation. The proposed BACE interface is designed to conformally wrap around the artery to accurately acquire the VE signals, thus providing critical insights into vasomotor states (Fig. 1a). The electrophysiological signals recorded by the interface reflect the electrical activity of autonomic nerve fibers innervating the vascular wall, capturing dynamic fluctuations in vasomotor tone regulated by sympathetic and parasympathetic inputs. Furthermore, based on its capability for electrical recording and stimulation, the BACE interface can facilitate the establishment of a bidirectional system for monitoring and regulating vasomotor dysfunction. Inspired by clinically validated neuromodulation strategies such as spinal cord stimulation, electrical pulses delivered through the interface are hypothesized to restore vasomotor balance by reactivating impaired neural pathways and reestablishing vascular smooth muscle responsiveness. The capability of bidirectional recording and modulation not only deepens the understanding of the bioelectrical mechanisms underlying vasomotor dynamics but also introduces promising therapeutic strategies for addressing vasomotor dysfunction.

### Design and characterization of the BACE interface

The interface was designed and fabricated with poly(3,4-ethylenedioxythiophene):poly(styrenesulfonate) (PEDOT:PSS)-coated gold (Au) electrodes serving as the contact with the vessel and polyimide (PI) functioning as the insulation and encapsulation layers (Fig. 1b, left and Fig. S1). The substrate of the interface, serving as the adhesive layer, is a free-standing and compliant film composed of SF/PU network. It functions similarly to an overcoat, where the adhesive outer edge makes contact with the vessel wall, thereby promoting stable mechanical coupling between the device and the artery. Silk fibroin is well-regarded for its mechanical flexibility, high biocompatibility, and ease of functionalization, making it ideal for geometrically reconfigurable and functionally versatile bioelectronic interfaces[24–26], while polyurethane is well-suited as a high-performance adhesive characterized by its internal structure with high bonding strength and fracture toughness[27,28]. Therefore, we fabricated SF/PU composites formed via mesoscopic reconstruction between the soft and hard segments of the PU backbone and the SF molecular chain (Fig. 1b, right). The developed network structure enhances mechanical performance, while the polar moieties in the composite enable tunable adhesion through intermolecular bonds (e.g., van der Waals forces, hydrogen bonding, capillary forces) and mechanical interlocking with tissue microstructures, adjustable by varying the PU proportion (Fig. 1c). Decreasing the ratio of SF/PU from 10/2 to 10/10 increased interfacial toughness from ~21 N/m to 139 N/m (Fig. S2). In addition, the incorporation of PU significantly softened the SF matrix, reducing its Young's modulus to a level approximately two orders of magnitude lower than that of bare PI (Fig. S3). The enhanced compliance of the SF layer also contributed to an overall reduction in the effective modulus

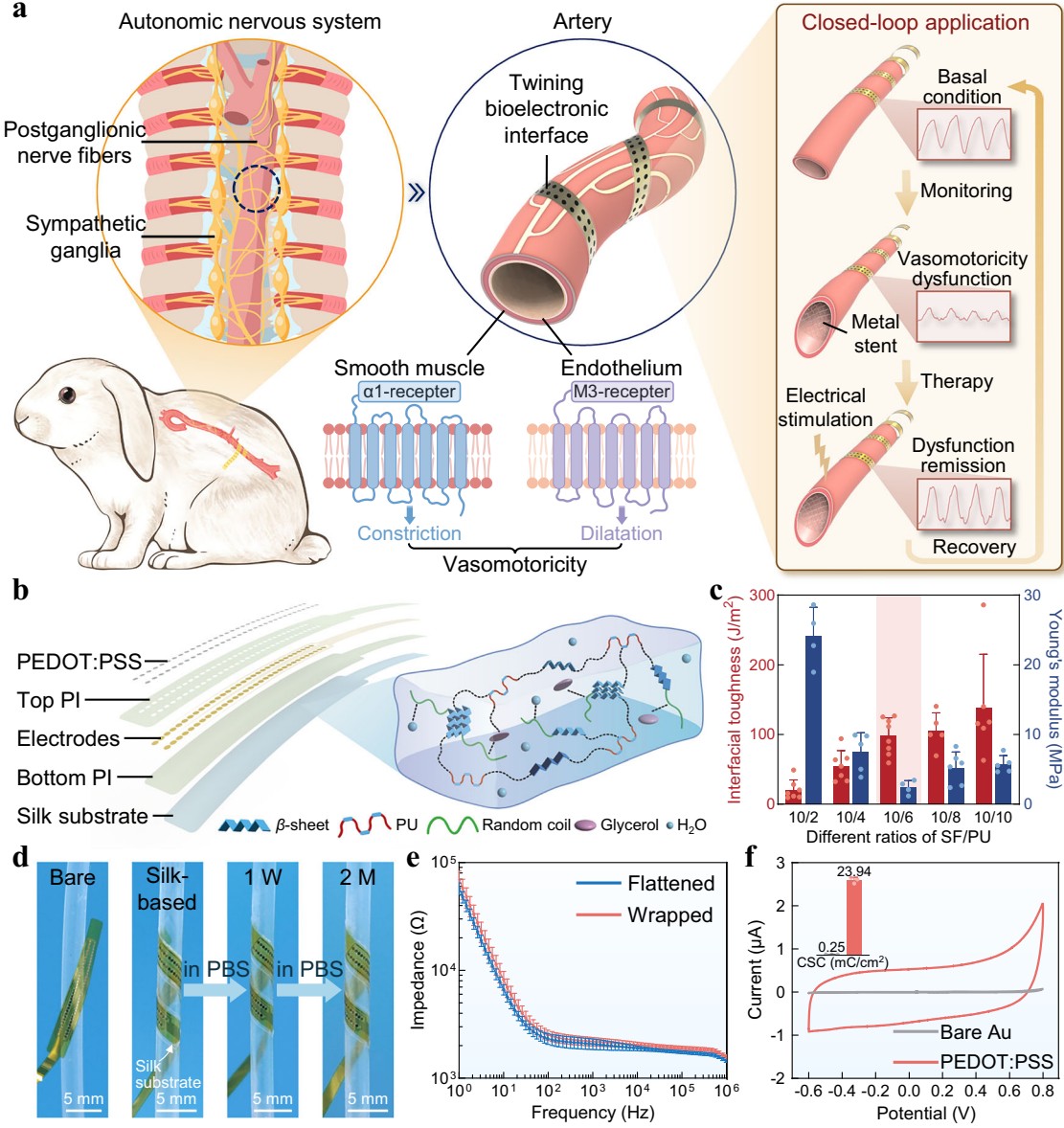

**Fig. 1 | Overview and characterization of the BACE interface. a** Schematic illustrations of the BACE interface wrapping around the artery to monitor the vasomotoricity, which is a multilevel coordinated biological regulatory process involving neural control by the autonomic nervous system and cellular regulation mediated by smooth muscle cells, endothelial cells, and receptor signaling pathways. The capabilities of the interface, encompassing both recording and stimulation, enable its application in bidirectional detection and regulation of vasomotor dysfunction. **b** Exploded-view schematic illustrating the functional layers of the BACE interface (left). Au electrodes are sandwiched between patterned PI serving as the top and bottom insulation. Surface modification of the electrodes is achieved via electroplated PEDOT:PSS. The silk substrate acts as a bioadhesive layer, with its structural features detailed on the right. **c** Interfacial toughness and Young's modulus of the SF/PU composites with different ratios. The selected ratio is shaded in pink. Data are presented as mean values ± SD. Sample sizes $n$ for interfacial toughness measurements were 7, 7, 8, 5, and 6, respectively; for Young's modulus measurements, they were 4, 5, 4, 6, and 5. All samples with the same SF/PU ratio were prepared using an identical protocol to ensure consistency. **d** The interface conformally wraps around the surface of a cylindrical tube immersed in 1x PBS for two months, with the silk substrate clearly indicated by the arrow. The cylinder, with an outer diameter of 3 mm, approximates the size of a rabbit's abdominal aorta. This demonstrates the mechanical compliance and adhesion compared to the interface based on bare PI. **e** Electrochemical impedance spectroscopy (EIS) of PEDOT:PSS-modified Au electrodes under flat and wrapping states. Data are presented as mean values ± SD, $n = 3$ individual interfaces in each group. **f** Cyclic voltammetry (CV) scans reveal an enhanced charge storage capacity (CSC) per unit area of PEDOT:PSS-coated Au electrodes compared to bare gold. The CV measurements were repeated independently across $n = 3$ interfaces, and a representative result is shown. The inset data are presented as mean values ± SD, $n = 3$ individual interfaces in each group.

of the BACE interface, offering improved mechanical conformity compared to PI-based substrates (Fig. S4). To balance adhesion strength and mechanical softness, we selected a 10/6 SF/PU ratio, which yielded a high interfacial toughness (~100 N/m) while maintaining a relatively low Young's modulus (<3 MPa). This formulation effectively enhances tissue adhesion without increasing the overall

bending stiffness of the interface, thus preserving its flexibility and minimizing mechanical mismatch with dynamic vascular surfaces (Fig. S5). The composite's mechanical properties enabled robust and seamless adhesion to a curved silicone tube surface with a bending diameter of 3 mm when immersed in phosphate-buffered saline (PBS) for two months, outperforming bare PI (Fig. 1d and Fig. S6). This long-

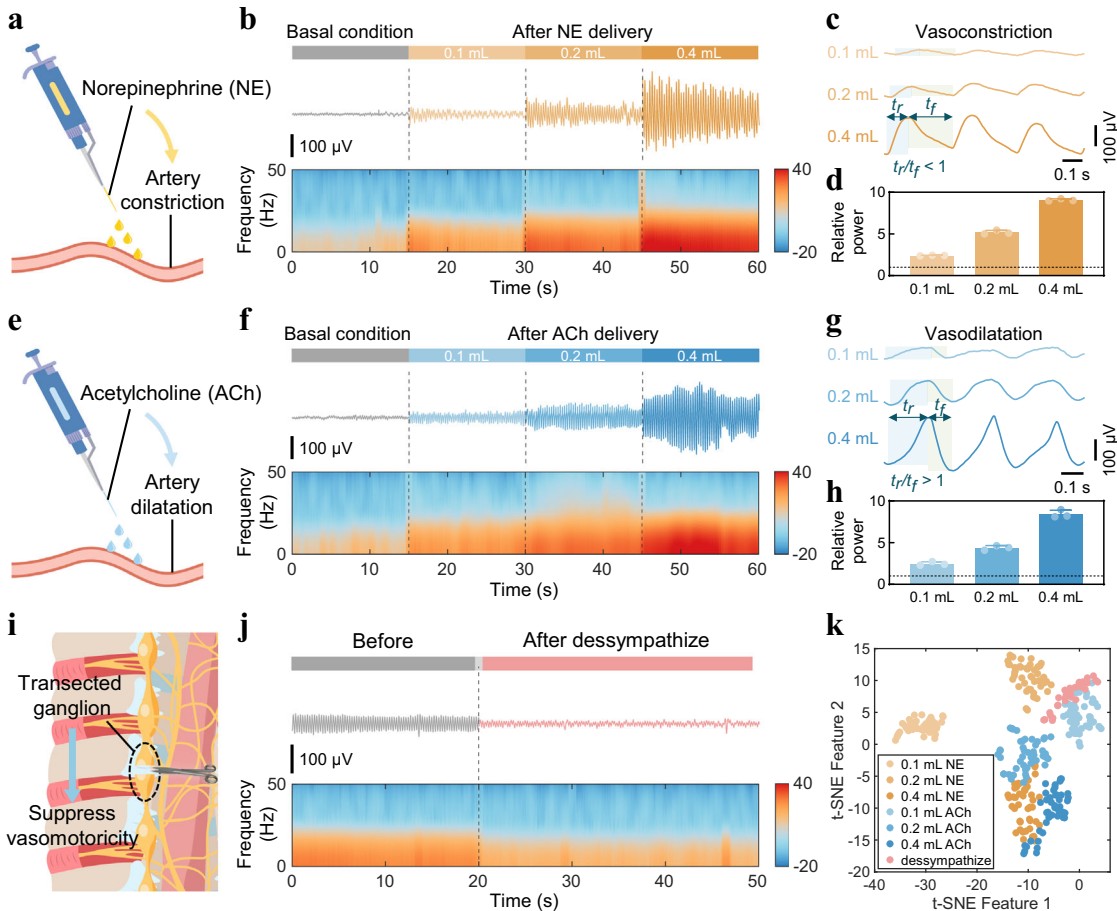

**Fig. 2 | Evaluation of precise vasomotor electrophysiology recording.**
**a** Schematic illustrating the exogenous administration of norepinephrine (NE) to the artery, inducing vasoconstriction. **b** Representative channel depicting electrical responses before and after the administration of different doses of NE (0.1, 0.2, and 0.4 mL) in an anaesthetized rabbit. **c**, Enlarged view of the VE signals following NE administration. **d** Relative amplitude of signals before and after the application of varying doses of NE. Data are presented as mean values ± SD from $n = 3$ individual rabbits in each group. **e** Schematic showing the exogenous delivery of acetylcholine (ACh) to the artery that leads to vasodilatation. **f** Representative channel showing electrical responses before and after the delivery of different doses of ACh (0.1, 0.2, and 0.4 mL) in an anaesthetized rabbit. **g** Enlarged view of the VE signals

after ACh delivery. **h**, Relative amplitude of signals before and after applying different doses of ACh. Data are presented as mean values ± SD from $n = 3$ individual rabbits in each group. **i** Schematic depicting sympathetic ganglion transection, resulting in vasomotor suppression. **j** Representative channel showing electrical responses before and after dessympathize in an anaesthetized rabbit.
**k** Visualization of VE signal feature space after dimensionality reduction via t-SNE, signifying varying vasomotor states induced by different interventions. The signals in (**b**, **f**, and **j**) are displayed as time-domain traces (top) and frequency-domain spectra (bottom). The time intervals between consecutive peaks and valleys are represented by $t_r$ and $t_f$.

term adhesion ensures that the mechanical stability does not pose a concern during extended recordings. Furthermore, we validated the BACE interface's resistance to mechanical disturbances through in vitro pulsatile flow simulations using a synthetic vessel system (Fig. S7).

Beyond mechanical robustness, the PEDOT:PSS-coated interface exhibited low impedance (~2 kΩ at 1 kHz) due to its large electrochemical surface area and high electronic and ionic conductivity. Impedance remained stable under morphological bending and distortion, demonstrating its reliable electrical performance (Fig. 1e). Moreover, PEDOT:PSS significantly increased the charge storage capacity (CSC) of the interface ($23.94 \, mC \, cm^{-2}$), compared to bare gold electrodes ($0.25 \, mC \, cm^{-2}$), due to its high volumetric capacitance, enabling enhanced stimulation efficiency for in vivo applications (Fig. 1f)[29–31]. In the in vivo environment, the effective adhesion of the interface to the surface of the abdominal aorta ensures reliable mechanical coupling and electrical pathways for robust connection with the vessels without restricting vascular expansion as well as inducing substantial motion-induced disruption (Figs. S8, 9), which are essential prerequisites for high-fidelity bioelectrical signal acquisition in dynamic physiological settings.

To further enhance its applicability toward long-term monitoring, we have developed a wireless BACE system integrating Bluetooth transmission and miniaturized implantable rechargeable battery. Preliminary tests have demonstrated its ability to reliably acquire and wirelessly transmit electrical signals with minimal distortion and high signal fidelity under in vitro and in vivo settings (Fig. S10).

**Evaluation of precise vasomotor electrophysiology recording**
To demonstrate the reliability and efficiency of VE signal recording, the BACE interfaces were implanted in direct contact with the exposed abdominal aorta of anesthetized rabbits to monitor vasomotor dynamics in vivo under various interventions. Pharmacologic intervention was first employed to induce significant enhancement in vasomotor function. Comparisons of drug application methods—dropwise addition to the arterial surface versus intravenous injection—revealed that intravenous delivery elicited a stronger response (Figs. S11, 12). Consequently, dropwise addition was selected to highlight the high sensitivity of the interface to subtle responses.

Given evidence that individual autonomic neurons can synthesize both norepinephrine (NE) and acetylcholine (ACh)[32], these

neurotransmitters were used to simulate vasomotor processes. NE, primarily acting on α-adrenoceptors in vascular smooth muscle, induces vasoconstriction upon activation. Exogenous NE delivery initiated the muscle contraction in the aorta (Fig. 2a). Conversely, ACh, acting on cholinergic receptors, stimulates nitric oxide production in endothelial cells, resulting in vasodilation under normal conditions (Fig. 2e)[33]. As both NE and ACh exert dose-dependent effects on vasomotor tone, the relationship between VE signals and vasomotor intensity was investigated by administering varying drug doses (Fig. S13). Baseline vascular electrical signals and those recorded under varying doses of NE (0.1, 0.2, and 0.4 mL) exhibited progressive increases in waveform amplitude and enhancement of low-frequency power in the spectrum, indicating dose-dependent intensification of vasoconstrictive activity (Fig. 2b). Similarly, graded administration of ACh resulted in a pronounced elevation of low-frequency power compared to baseline, consistent with enhanced vasodilatory responses (Fig. 2f). Both NE and ACh significantly amplified arterial vasomotion, as evidenced by increased vascular electrical activity. Close examination of the corresponding signals during vasoconstriction and vasodilatation further revealed distinct oscillatory patterns specific to vasoconstriction and vasodilation states (Fig. 2c, g). Notably, the ratio of rising edge duration to falling edge duration within individual signal cycles displayed consistent and state-dependent variations, serving as a distinguishing feature of different vasomotor dynamics. The dominant frequency bands are attributable to electrophysiological responses related to the sympathetic nerve activity that regulates vascular tone. Frequency-domain analysis excluded potential confounding effects from low-frequency components, confirming that the observed changes in signal characteristics primarily reflect physiological alterations rather than baseline oscillations (Fig. S14). Quantitative analyses confirmed a strong correlation between electrical response magnitude and the extent of vasomotion, with relative signal amplitudes before and after drug administration increasing significantly with higher doses (Fig. 2d, h).

On the other hand, the sympathetic ganglion plays a critical role in regulating the autonomic functions of abdominal organs and celiac arteries. Therefore, surgical transection of the sympathetic ganglion disrupts the sympathetic nerve system, resulting in a loss of sympathetic tone in the abdominal aorta. This manifests as transient vasomotor dysfunction, especially impaired vasoconstriction (Fig. 2i). Denervation leads to a significant reduction in VE activity, while compensatory mechanisms involving intact ganglia and circuits help maintain partial arterial function, allowing for the detection of weak electrical signals (Fig. 2j and Figs. S15, 16). Following sympathetic ganglion denervation, electrical stimulation of the abdominal aorta with 100, 300, and 500 μA currents elicited a modest recovery of VE signals compared to the post-denervation baseline (Fig. S17). However, the signal amplitudes remained lower than those observed prior to nerve transection, highlighting both the therapeutic potential and the physiological limitations of stimulation in the context of disrupted autonomic control. In addition, in arterial occlusion models, the BACE interface successfully detected different signal characteristics under proximal and distal blockade, confirming its sensitivity to acute vasomotor conduction (Figs. S18, 19). These results further demonstrate the interface's ability to monitor localized vascular dysfunction under pathophysiological conditions.

To intuitively differentiate vasomotor states encoded by recorded vascular bioelectrical activities, we employed t-distributed stochastic neighbor embedding (t-SNE) for dimensionality reduction of VE signal features. Compared to configurations with different axial spatial resolution, the 64-channel BACE interface with sufficient resolution enabled clearer separation of activity patterns induced by different interventions, highlighting its comprehensive recording capability (Fig. 2k and Figs. S20–22). These results demonstrate that seamless wrapping of the BACE interface around the aorta enables precise electrophysiological recording of vasomotor dynamics, which has practical potential to quantify the degree of vasomotor states and further monitor and forecast the severity of vasomotor dysfunction.

## Stent-induced vasomotor dysfunction monitoring

The implanted metal stent may compromise the integrity of the vascular endothelium and the release of vasodilatory factors such as nitric oxide, thereby disrupting normal vasomotor function. Building upon the successful validation of the BACE interface for precise recording of vasomotor activity, we sought to further explore its potential for monitoring vasomotor dysfunction in vessels implanted with a stent. The vascular disease model was established by deploying a metal stent in the midsection of the rabbit abdominal aorta (Fig. 3a). The BACE interface conformed seamlessly to the aorta, ensuring intimate contact and reducing tissue-electrode impedance, with all 64 recording sites demonstrating impedance values of $6.77 \pm 2.13 \, k\Omega$ (Fig. 3b). Meanwhile, the implanted BACE interfaces were able to record valuable physiological information associated with autonomic nervous activity, yielding low baseline noise ($2.63 \pm 0.52 \, \mu V$) and stable signal amplitude (Fig. 3c).

While DSA and ultrasound imaging are widely recognized as gold standards for clinical vascular assessment, they primarily provide structural and hemodynamic data without direct evaluation of vasomotor function. Although angiographic (i) and ultrasound (ii) images revealed a gradual, modest narrowing of the distal aorta from immediately post-stent implantation to one month later (Fig. 3d–f and Fig. S23), they did not offer direct insights into changes in vasomotor function. In contrast, our multi-channel BACE interface enabled high-resolution tracking of electrophysiological changes in the distal aorta, directly reflecting vasomotor status. Heatmaps of electrical activity and representative signal waveforms from the distal aorta demonstrated a decrease in both power and amplitude, providing a direct visualization of the progressive decline in distal vasomotor function induced by stent implantation (Fig. 3d–f (iii) and Figs. S24–26). Further feature analyses confirmed that this attenuation was statistically significant both immediately following stent implantation and one month thereafter (Fig. 3g, top and Fig. S27). Vasomotoricity is closely related to vascular stiffness and elasticity, as impaired elasticity and increased stiffness typically reduce the vessel's ability to undergo proper constriction and dilation. To validate the reliability of our BACE interface in assessing vasomotor dysfunction, we incorporated the arterial stiffness parameter $\beta$ for comparative analysis with electrophysiology (more details in "Methods" section)[34]. As duration increased, the stiffness parameter of the distal aorta increased, indicating a decline in vascular elasticity and compliance in response to blood flow pressure (Fig. 3g, bottom). The comparison of bioelectrical analysis with clinical gold standards at three distinct time points confirmed that our BACE interface provides a robust method for identifying abnormalities in vasomotor functions following interventional therapy.

Having characterized the suppression of VE activity due to stent implantation, we then investigated whether vasoactive drugs could restore normal vasomotor function. Specifically, we compared the changes in signal amplitude before and after NE delivery in arteries with normal vasomotor function and those with stent implantation (Fig. 3h). The results revealed a failure of physiological regulatory drugs to restore normal function, underscoring the need for alternative approaches to regulate vasomotor dysfunction.

Beyond its utility in identifying vasomotor impairment following stent implantation, the BACE interface also demonstrated robust performance in capturing electrophysiological alterations associated with other aortic pathologies, such as in the abdominal aortic aneurysm (AAA) model (Fig. S28). This expands the scope of the interface to support mechanistic investigations into AAA progression and its impact on neurovascular regulation. Furthermore, the BACE system is not confined to large central arteries; its conformable design and high

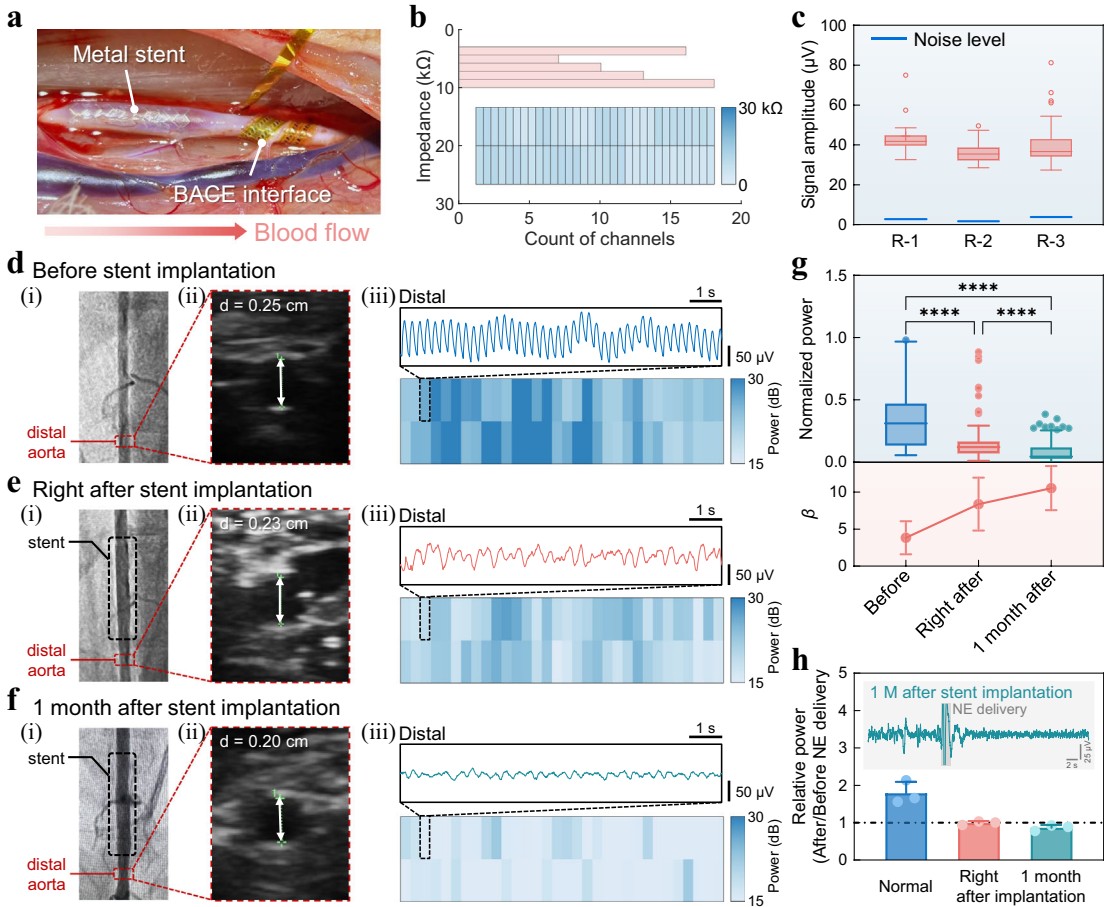

**Fig. 3 | Stent-induced vasomotor dysfunction monitoring. a** Photograph of the experimental setup showing the BACE interface implanted in the distal segment of the aorta, relative to the metal stent. **b** Histogram of electrode impedance (measured at 1 kHz frequency) for the 64-channel interface in (**a**). Inset: Spatial distribution of the impedance magnitudes corresponding to the actual channel positions. **c** Quantification of recorded signal amplitudes under basal conditions compared with baseline noise (indicated by blue line) for three individual implanted interfaces on three rabbits: $n = 61$ (R-1), 59 (R-2), and 63 (R-3) channels. Multimodal imaging and electrophysiological recordings at three critical time points: before stent implantation (**d**), right after stent implantation (**e**), and 1-month post-implantation (**f**). **i** DSA images showing the abdominal aorta of the rabbit; (**ii**) ultrasound cross-sectional views of the distal aorta with measured diameters; (**iii**) Representative traces of vascular electrical activity (top) and power spatial distributions across 64 channels (bottom) recorded from the distal aorta using the BACE interface. **g** Comparison of normalized power of the VE signals acquired from the distal segment of the aorta at the three time points from $n = 3$ rabbits (top). $p$ values for power comparison: $p = 3.0474 \times 10^{-10}$ (Before vs. Right after), $p = 1.4493 \times 10^{-34}$ (Before vs. 1 month after), $p = 1.0702 \times 10^{-7}$ (Right after vs. 1 month after). (*$p < 0.05$, **$p < 0.01$, ***$p < 0.001$, ****$p < 0.0001$, Kruskal-Wallis test with Dunn's post hoc test). Arterial stiffness parameters $\beta$ at the same time points were calculated for validation (bottom), and data are presented as mean values $\pm$ SD from $n = 3$ individual rabbits. **h** Relative power of VE signals from normal and stent-implanted (both immediately and one month later) rabbits before and after treatment with NE. Data are presented as mean values $\pm$ SD from $n = 3$ individual rabbits. Inset: A representative waveform recorded from the aorta 1 month after implantation. Box plots in (**c**) and (**g**) (top) depict the data median (center line), upper and lower quartiles (box bounds), 1.5 times the interquartile range (whiskers), and outlier values beyond this range (circles).

signal sensitivity enable reliable monitoring of vasomotor dysfunction in smaller peripheral arteries as well. By providing access to electrophysiological signatures from anatomically accessible, distal vascular sites, the interface offers meaningful surrogate insights into central aortic physiology (Fig. S29). These capabilities collectively highlight the scalability and translational relevance of our platform, suggesting potential for broader clinical applications across a range of vascular territories and disease contexts.

**Bidirectional modulation for vasomotor dysfunction**
In addition to its capacity for monitoring vasomotor dysfunction, the recording and stimulation performance of the BACE interface facilitates the demonstration of a bidirectional regulation system, highlighting its potential as a therapeutic platform. The interface, positioned at the distal end of the artery, can monitor vasomotor status and detect dysfunction by identifying signal patterns that

deviate from the healthy state, which refers to the baseline electrophysiological pattern recorded from non-stented rabbits under basal physiological conditions characterized by relatively high and stable amplitude and preserved responsiveness to sympathetic stimulation. This dysfunction then triggers the interface at the stent site to apply electrical stimulation pulses, following parameter optimization, to regulate vasomotor function (Fig. 4a). We simultaneously implanted two interfaces on the rabbit aorta: one at the distal segment for monitoring and the other at the stent site for modulation (Fig. 4b). Prior to initiating the electrical modulation, we conducted a systematic evaluation of stimulus pulses at varying current amplitudes, integrating analyses of time-domain waveforms, quantitative power metrics, and physiological safety indicators such as heart rate variability. While 100 μA produced only marginal changes in vascular electrophysiological activity, 300 μA generated the most robust and stable augmentation of responses without compromising cardiac rhythm,

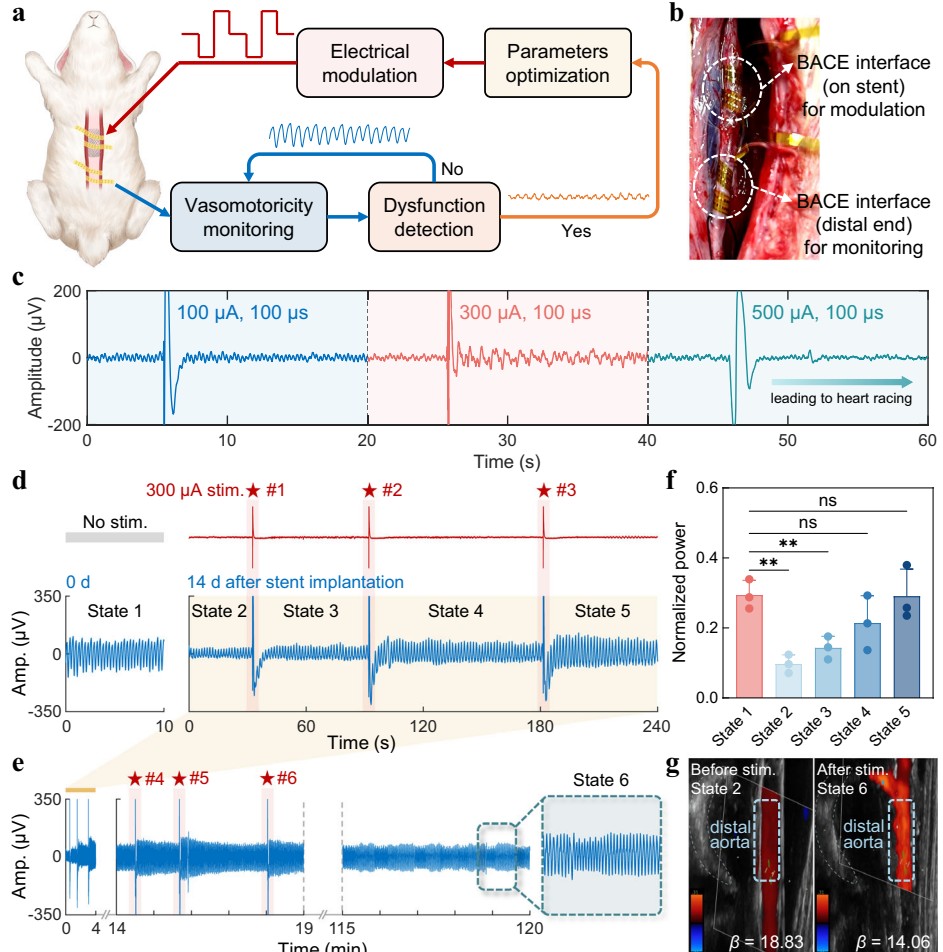

**Fig. 4 | Bidirectional modulation of vasomotor dysfunction. a** Block diagram of the bidirectional system designed for detecting and modulating vasomotor dysfunction. **b** Photograph showing the implantation sites and functional roles of the two interfaces. **c** Optimization and selection of stimulation pulse current amplitude. **d** VE signals recorded by the BACE interface at the distal segment of the aorta in response to a series of stimulations with a charge-balanced current pulse (pulse amplitude: 300 μA, pulse width: 100 μs) immediately and 14 days after stent implantation. The timing of stimulation is indicated by red stars. **e** Extended monitoring of vasomotoricity over a prolonged period based on (**d**), incorporating three additional current stimulations. A close-up view highlights the steady-state VE

signal following the cessation of stimulation. **f** Comparative analysis of normalized power across five states annotated in (**d**) from $n = 3$ rabbits. Data are presented as mean values ± SD. $p$ values for power comparison: $p = 0.0011$ (State 1 vs. State 2), $p = 0.0045$ (State 1 vs. State 3), $p = 0.0716$ (State 1 vs. State 4), $p = 0.9173$ (State 1 vs. State 5). (ns: $p > 0.05$, *$p < 0.05$, **$p < 0.01$, ***$p < 0.001$, ****$p < 0.0001$, One-way repeated measures analysis of variance). **g** Ultrasound color-flow imaging of the distal aorta, along with corresponding arterial stiffness parameters $\beta$ before and after stimulation (with the corresponding states annotated), demonstrating the effectiveness of electrical modulation. Stim. in (**d**) and (**g**) denotes stimulation.

whereas 500 μA precipitated arrhythmogenic events (Fig. 4c and Figs. S30, 31). Consequently, 300 μA was selected as the optimal current for subsequent experiments.

Considering the peak phase of intimal hyperplasia and the effect of the accumulation of sirolimus released by the drug-eluting stents we used[35], which aggravates vasomotor dysfunction, the intermittent electrical pulses were applied 14 days after stent implantation. A progressive increase in the amplitude of bioelectrical signals from the distal aorta can be observed following each stimulation, representing a recovery or enhancement of the overall vasomotor competence after stent implantation (Fig. 4d and Fig. S32). This increase is likely attributable to the activation of vasomotor regulatory pathways, including ion channel-mediated signaling and sympathetic nerve terminal involvement, which are known to play essential roles in vasomotor tone control[36,37]. Extended continuous monitoring of the distal aorta revealed that, after several electrical stimulations, the signal stabilized, suggesting the sustained therapeutic effects of electrical stimulation (Fig. 4e). To quantitatively assess the vasomotor modulation capabilities, we selected several data points from Fig. 4d for statistical

analysis of the VE responses. Our analysis indicated that electrical stimulation significantly contributed to the recovery of signal power. After two stimulations, the vasomotor function, as represented by the electrophysiological signals, returned to its baseline state (Fig. 4f). This sustained effect may reflect the prolonged activation of endothelial and smooth muscle cell communication via intercellular gap junctions, allowing the electrical signal to propagate distally and maintain vascular tone modulation beyond the stimulation period. These findings were further validated by ultrasound measurements and corresponding arterial stiffness indicators. Figure 4g presents longitudinal ultrasound images of the abdominal aorta before and after stimulation, with measurements of the maximum and minimum distal arterial diameter during systolic and diastolic periods. The calculated arterial stiffness parameter $\beta$ revealed a decrease from 18.83 before stimulation to 14.06 after stimulation, indicating reduced stiffness and improved elasticity in the local arterial wall. Notably, our current results demonstrate that the electrophysiological modulation induced by stimulation remains stable for up to 5 h, highlighting the prolonged duration of neural regulation of vascular activity (Fig. S33). Moreover,

the arterial stiffness parameter $\beta$ assessed at approximately 5 h after the final stimulation still showed significant improvement compared to the pre-stimulation baseline, providing strong evidence that the effects of electrical stimulation persist well beyond the immediate intervention window. These results suggest that electrical stimulation may enhance electrical signal conduction in dysfunctional vessels, thereby improving vascular elasticity and compliance. This presents a promising therapeutic strategy for preventing the progressive suppression of vasomotor responsiveness in the distal aorta after stent deployment.

## Discussion

Vasomotor dysfunction results from a disruption in the delicate balance among endothelial function, smooth muscle contractility, and neurovascular signaling, all of which are essential for maintaining vascular integrity. This disruption is frequently relevant in the context of stent implantation, which can give rise to distal vascular complications associated with vasomotor dysfunction. Nevertheless, the underlying mechanisms, especially the connection between these abnormalities and vascular electrical signaling, remain inadequately understood. The lack of advanced technology for high-quality VE recording has been a significant barrier in studying vasomotor dysfunction. In this context, the BACE device introduced here demonstrates a promising route towards robust and high-performance flexible bioelectronics that can conformally wrap around blood vessels and accurately monitor vascular electrical activity closely linked to vasomotoricity. Beyond diagnostics, the integration of localized stimulation capabilities within the BACE platform offers potential therapeutic applications. By modulating vasomotor activity through targeted electrical stimulation, our approach holds promise as a therapeutic platform for managing vasomotor dysfunctions associated with a wide range of vascular diseases in clinical settings. This dual functionality positions the BACE device as a versatile tool poised to advance both the study and treatment of vascular disorders.

The interplay between vasomotor function and hemodynamics is both complex and bidirectional. While vasomotor reactivity is a key mechanism in modulating hemodynamics to adapt to the metabolic demands of the body, hemodynamic forces can also influence vasomotoricity through mechanotransduction pathways, myogenic regulation, and neurovascular coupling[38–40]. Previous studies have confirmed hemodynamic changes, such as pulse wave and pulse rate, during vasoconstriction and vasodilation induced by pharmacological interventions[21,22]. However, electrophysiology plays an integral role in the dynamic regulation of vascular function. The coordination of vasomotor activity depends on the propagation of electrical signals between smooth muscle cells and endothelial cells, a process known as the conducted vasomotor response, which is essential for maintaining vascular homeostasis and ensuring efficient blood flow distribution[41]. Therefore, in contrast to existing clinical imaging modalities and studies on vascular mechanical sensors (Table S1), our research extends previous findings by offering complementary insights into the dynamic regulation of vasomotor function from a bioelectrophysiological perspective, thereby broadening the scope of vascular physiology and pathology studies.

Although this study primarily focuses on the macroscopic sensing of vascular electrophysiological signals using the BACE interface, it opens several important directions for future exploration. Integrating this approach with cellular and molecular-level investigations will further elucidate the fundamental mechanisms underlying vasomotor activity. Notably, sympathetic ganglion transection was employed in this work to disrupt sympathetic input and examine its role in vasoconstriction; however, due to technical limitations, parasympathetic ganglia were not clearly transected, and their contribution to VE signal modulation remains to be fully validated. Moving forward, the

development of a fully implantable wireless BACE system presents a critical next step. Such a system—capable of continuous in vivo monitoring through integrated wireless power delivery and data transmission—would eliminate the need for external wiring, thereby reducing the risk of infection and mechanical disturbance. Beyond improving recording stability and safety, a wireless platform would facilitate long-term tracking of vasomotor dynamics in freely moving subjects, offering clinically valuable insights into disease progression and enabling early detection and personalized management of vascular disorders such as aortic aneurysms and arterial dissection. Ultimately, the engineering advancement of intelligent, minimally invasive bioelectronic interfaces will lay the groundwork for closed-loop vascular systems that can non-invasively monitor, diagnose, and therapeutically respond to abnormal vasomotor conditions in real time, driving significant advancements in precision vascular medicine.

## Methods

### Preparation of silk solution

*Bombyx mori* silk fibroin was prepared following the established protocol. First, *Bombyx mori* cocoons were boiled in a 0.02 M $Na_2CO_3$ (Sigma-Aldrich, USA) solution for 30 min and then rinsed in deionized water for $3 \times 30$ min to remove the sericin. After drying for over 12 h, the dry silk fibers were dissolved in a 9.3 M LiBr (Sigma-Aldrich, USA) solution at 60 °C for about 4 h. The solution was subsequently dialyzed in deionized water for 48 h using Slide-A-Lyzer dialysis cassettes (Molecular weight cut-off, MWCO 3.5 kDa, Pierce, USA). The solution was then centrifuged at ‑24,000 $g$ for $2 \times 20$ min to obtain the purified silk solution. The final concentration of the silk solution was determined by measuring the dry weight of a volume of the solution.

### Preparation of the SF/PU composites

All chemicals were used as received without further purification. The purified silk fibroin solution was first mixed with glycerol (99.5%, Aladdin, China) as a plasticizer at a 5% volume ratio. Subsequently, the water-based polyurethane resin (Guanzhi New Material Technology Co., Ltd., China) was added in varying volume ratios ranging from 20% to 100% to the silk solution. The resulting mixture was then sonicated at room temperature for 1 min to ensure uniform dispersion. The prepared solution was subsequently cast onto a clean polydimethylsiloxane (PDMS) substrate using a pipette. The samples were allowed to cure for at least 2 h at room temperature. The resulting thickness of the composites can vary between 10 μm and 100 μm, depending on the volume of the solution applied.

### Preparation of the polyimide-based interface

The polyimide-based interface was fabricated using standard microfabrication techniques[42]. Initially, a 7 μm-thick bottom layer of PI (JA-101, Jingai Microelectronics Co., Ltd., China) was spin-coated onto a silicon wafer. The electrodes and interconnectors were then patterned via photolithography, followed by electron beam evaporation and a lift-off process, resulting in a metal layer of 15 nm-thick chromium (Cr) and 350 nm-thick gold (Au). Similarly, the bonding pads were formed with layers of 5 nm-thick Cr/100 nm-thick Nickel (Ni)/50 nm-thick Au to serve as solder joints for subsequent attachment to a back-end printed circuit board (PCB). A 13 μm-thick top layer of PI was spin-coated to function as the insulation layer. An aluminum layer was sputtered as the hard mask for etching. Through UV photolithography, aluminum corrosion, and oxygen plasma dry etching, opening areas of the electrodes and bonding pads were defined. The wafer underwent reactive ion etching (RIE) to expose the recording sites and pads, followed by aluminum wet etching to remove the hard mask. The free-standing interface, featuring a sandwich structure with metal contacts embedded within flexible sheets of PI, was released from the silicon wafer via wet etching. Reflow soldering was employed to bond the interface to

the customized PCB (Shenzhen Station Electronic Technology Co., Ltd.), enabling simultaneous addressing and acquisition of 64 channels.

## Surface modification of the interface

To obtain optimal electrical properties characterized by low impedance and high charge storage capacity, the electrodes of the interface were modified with PEDOT:PSS. Electrochemical modification was performed using an electrochemical workstation (CHI660e, CH Instruments, China). The electroplating solution consisted of 0.01 M ethylenedioxythiophene (EDOT, Sigma-Aldrich, USA) and 0.1 M PSS (Sigma-Aldrich, USA). A three-electrode configuration was employed, with the recording sites serving as the working electrode, a platinum wire as the counter electrode, and an Ag/AgCl electrode as the reference electrode. Electroplating was conducted in potentiostatic mode at 3 V. Following modification, the interfaces were immersed in deionized water for 24 h to remove any residual electrolyte, and then dried for subsequent use.

## Mechanical characterization

All the mechanical tests were conducted using a CMT4204 tensile testing machine (SUST, China) equipped with a 20-N load cell. To measure interfacial toughness, adhered samples with widths of 20 mm were prepared and tested following the standard 180° peel test procedure (ASTM F2256). All tests were performed at a constant peeling speed of 10 mm min⁻¹. The measured force reached a plateau as the peeling process entered the steady state. Interfacial toughness was determined by dividing twice the plateau force by the width of the tissue sample[43].

The tensile properties of the SF/PU composites were measured following the standard tensile test protocol (ASTM D412). Samples with sizes of 40 mm in length and 10 mm in width were prepared and equilibrated in 1× PBS at room temperature for 24 h prior to testing. All the tests were conducted at a constant tensile speed of 5 mm min⁻¹. The effective Young's modulus was determined from the initial slope of the stress-strain curve.

## Electrochemical characterization

To evaluate the electrical performance of the interface after electrochemical modification, electrochemical impedance spectroscopy (EIS) and cyclic voltammetry (CV) measurements were performed using an electrochemical workstation (CHI660e, CH Instruments Inc., China). The setup for testing was the conventional three-electrode configuration in 1x PBS (pH 7.2–7.4) at room temperature, with the test sample as the working electrode, an Ag/AgCl electrode as the reference electrode and a Pt wire as the counter electrode. EIS measurements were carried out in the Bode model with a frequency range of 1 Hz to 1 MHz, using a sinusoidal wave with an AC voltage amplitude of 5 mV. CV scans were performed over the potential range of −0.6 V to 0.8 V at a scan rate of 0.05 V s⁻¹. The CSC was calculated from the CV curve by integrating the current over the potential range. To determine the areal capacity, the calculated CSC was normalized by the electrode area.

## In vivo experiments and signal recording

All animal procedures were performed in accordance with the Guide for the Care and Use of Laboratory Animals published by the U.S. National Institutes of Health. The experimental protocol was approved by the institutional ethics committee of Huashan Hospital, Fudan University (Approval number: 202410031S). The experimental New Zealand White rabbits, sourced from FMC Laboratory Animal Technology Co., LTD. (Zhejiang, China), were 18 weeks old, male, and weighed 3–3.5 kg. Anesthesia was induced via marginal ear vein injection of Zoletil 50 (0.1 mL/kg), and the rabbits were securely positioned in the supine orientation for subsequent procedures.

To perform the transection of the sympathetic ganglion, the precise location of the ganglia was first identified. Sympathetic ganglia are primarily situated in two regions: the sympathetic chain ganglia and the prevertebral (or collateral) ganglia[44]. The prevertebral ganglia consist of clusters of sympathetic nerve cell bodies located in the abdominal cavity, anterior to the vertebral column, and close to major arteries. Following the opening of the abdominal cavity, the celiac ganglion and superior mesenteric ganglion were readily identified near the celiac trunk and superior mesenteric artery. These ganglia, arranged in a chain-like formation parallel to the spinal cord, of which the sympathetic ganglion was carefully separated and transected.

For the implantation of the stents, the right femoral skin of the rabbit was exposed, sterilized, and a longitudinal incision of approximately 25–30 mm was made. Arterial access was established via a 20 G needle puncture (Terumo, RS*A70K10SQ, Japan) of the right femoral artery, followed by the insertion of a 4-Fr introducer sheath (Terumo, RS*A70K10SQ, Japan). Abdominal aortic angiography was performed using 4F guiding catheters under digital subtraction angiography (GE Medical, INNOVA2100, USA) through manual syringe injection of contrast medium. Subsequently, a midline abdominal incision of approximately 60 mm was made to fully expose the abdominal aorta. The diameter of the abdominal aorta was measured via angiography, and the stents were selected and expanded to 1.1–1.2 times the diameter of the target vessel. In this study, stents with a diameter of 3 mm (Firehawk, MicroPort, China) were used. The stents were implanted through a 4F sheath and positioned in the abdominal aorta below the bilateral renal artery. During the expansion process, the stents were initially pressurized to 3 atm for 10 s, followed by an incremental increase at a rate of 1 atm per second until the target pressure (typically 8–12 atm) was reached. The final pressure was maintained for 10–20 s before deflating the balloon and withdrawing it. After implantation, the arterial sheath was removed, and the femoral artery was ligated. All incisions were sterilized and sutured. One month after stent implantation, 0.2 mL of norepinephrine (NE) was exogenously applied to the adventitia of the abdominal aorta in a manner consistent with that in Fig. 2, with the aim to assess whether NE could modulate vascular electrophysiological activity following long-term stent implantation.

Vascular electrophysiology recordings were performed by conformally wrapping the BACE interface around the abdominal aorta and connecting the implant to a multichannel data acquisition system CereCube NSP8 (Neuroxess Co., Ltd., China). Data were collected, amplified, and digitalized with a sampling frequency of 4 kHz. The reference wire was embedded subcutaneously near the surgical incision to ensure accurate signal acquisition.

## Ultrasound imaging and evaluation of arterial stiffness parameter

The technique for measuring stiffness parameter $\beta$ was performed using ultrasound devices and sphygmomanometers. Briefly, the distal aorta of the rabbits, located about 2 cm above the bifurcation, was measured in the supine position. The ultrasound devices used in this study are equipped with a phase-locked echo-tracking system with a real-time scanner and a high-resolution linear array probe (ProSound II, SSD6500, α7, α10, Hitachi-Aloka Medical Ltd., Japan). Meanwhile, blood pressure measurements at the forelimb artery of the rabbits were obtained using a sphygmomanometer, which was then used to calculate stiffness-related indices. Arterial stiffness parameter $\beta$ is obtained from the changes in arterial diameter during the cardiac cycle and blood pressure of the local artery at the measurement site, which is defined as follows[45]:

$$\beta = \ln\left(\frac{P_s}{P_d}\right)\bigg/\left(\frac{D_s - D_d}{D_d}\right) \tag{1}$$

where $P_s$ and $P_d$ represent systolic and diastolic blood pressure, respectively. $D_s$ and $D_d$ denote the maximum and minimum of arterial diameter during systolic and diastolic period. The measurements for arterial diameter using ultrasound were performed by an experienced sonographer, with each result measured at least three times. Notably, it is an important technique for an examiner to put a probe on the surface as gently as possible so as not to restrict the movement of the arterial wall during the cardiac cycle. A higher $\beta$ value indicates increased arterial stiffness. Arterial stiffness parameter is also a marker of atherosclerotic vascular damage and an independent predictor of development of risk of cardiovascular disease[46].

### Electrical stimulation of the artery

The BACE interface was implanted in the middle segment of the artery for electrical stimulation. Charge-balanced, biphasic symmetric current pulses (cathodal first, 100 µs pulse width) were delivered using a multichannel recording and stimulation system CereCube NSPS8 (Neuroxess Co., Ltd., China). The range of current amplitudes for stimulation was partially guided by previous research on peripheral nerve stimulation[47–52]. A reference wire was positioned subcutaneously near the surgical incision to ensure precise signal acquisition. Electrical signals from both the middle and distal segments of the artery were recorded simultaneously at a sampling rate of 30 kHz and bandpass filtered between 1 Hz and 7.5 kHz.

### Vascular electrophysiology analysis

Vascular electrophysiology analysis was performed in MATLAB R2021a (MathWorks, Inc., USA). The recorded electrical signals were imported, and corrupted channels were removed. Unless otherwise specified, notch filters were applied to remove mains noise. To reduce common-mode noise in the multi-channel data, the common average reference (CAR) technique was utilized. The processed data were then bandpass filtered using a fourth-order Butterworth filter with cut-off frequencies of 0.5 Hz and 50 Hz. The root-mean-square (RMS) values were computed using a 100-ms sliding window.

With regard to the computation of relative power, the term relative refers to a within-subject comparison before and after a specific intervention or stimulation (e.g., pharmacological administration or surgical manipulation). For each subject, we identified two stable segments of signal-one preceding and one following the intervention. Within each segment, the signal's power was estimated by computing the power spectral density over the frequency range of 0.5–50 Hz using a sliding window approach with a window length of 100 ms and a 50% overlap. The power of each window was computed and then averaged across the entire segment to obtain a representative power value for both pre- and post-intervention periods. The relative power was defined as the ratio of the post-intervention average power to the pre-intervention average power for each subject. This approach yields a dimensionless quantity (unitless ratio), allowing us to normalize inter-subject variability and focus on the proportional changes in signal power induced by the experimental manipulation.

t-SNE, an unsupervised dimensionality reduction technique, was employed to preserve the local structure of the data by minimizing the divergence between probability distributions of pairwise similarities in both high-dimensional and lower-dimensional spaces. This approach was used to illustrate the high-resolution recording capability of the BACE interface to evaluate various degrees of vasomotoricity induced by different interventions. The ratios of waveform rhythm (Fig. 2c, g), along with time-domain and frequency-domain characteristics of the signals before and after the intervention, were calculated and organized into feature vectors, which served as the input for the analysis. The results were visualized in a two-dimensional space based on the first two principal components.

### Statistical analysis

All data were presented as mean values ± standard deviation (SD), unless otherwise specified. The experiments involving statistical tests were conducted with a minimum of $n = 3$ replicates. Statistical analyses were conducted using GraphPad Prism version 10.1.2 (GraphPad Software Inc., USA). The significance threshold was denoted as * for $p \leq 0.05$, ** for $p \leq 0.01$, *** for $p \leq 0.001$, and **** for $p \leq 0.0001$, respectively. Non-significant results were indicated as ns (not significant).

### Ethics

Every experiment involving animals has been carried out following a protocol approved by an ethical commission (Approval number: 202410031S).

### Reporting summary

Further information on research design is available in the Nature Portfolio Reporting Summary linked to this article.

## Data availability

All data supporting the findings of this study are available within the article and its supplementary files. Any additional requests for information can be directed to and will be fulfilled by the corresponding authors. Source data are provided with this paper. The large electrophysiological datasets used in this study are available in the FigShare repository under accession code [https://doi.org/10.6084/m9.figshare.29559218]. Source data are provided with this paper.

## Code availability

The customized MATLAB scripts for electrophysiology analysis are available from Zenodo (https://doi.org/10.5281/zenodo.16017079)[53].

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

## Acknowledgements

This work was partially supported by the Youth Innovation Promotion Association for Excellent Members, CAS (Grant No. Y2023070, Z.Z.), Shanghai Rising-Star Program (Grant No. 22QA1410900, Z.Z.), National Natural Science Foundation of China (Grant Nos. 52371249, J.T.; L2324226, B.Y.), Key Research Program of Frontier Sciences, CAS (Grant No. ZDBS-LY-JSC024, T.H.T.), Shanghai Pilot Program for Basic Research-Chinese Academy of Science, Shanghai Branch (Grant No. JCYJ-SHFY-2022-01, T.H.T.), Natural Science Foundation of Shanghai

(Grant No. 24ZR1407900, W.F.), Shanghai Municipal Science and Technology Major Project (Grant No. 2021SHZDZX, L.S.).

## Author contributions

X.W. and W.F. contributed equally to this work. J.T., Z.Z., T.H.T., X.W., W.F. and B.Y. conceived the idea. Z.Z., J.T., T.H.T., X.W. and W.F. designed the experiments. Z.Z., X.W., J.T., and E.Z. designed the device. X.W. and Y.L. fabricated and prepared the interface. X.W., W.F., Z.Z., J.T., L.C., L.S. and X.L.W. performed the experiments. X.W., Z.Z., W.F., J.T., and L.C. analyzed the data. X.W., W.F., Z.Z., J.T. and T.H.T. wrote and reviewed the paper. All authors discussed the results and provided comments for the manuscript.

## Competing interests

The authors declare no competing interests.
