## [Transparent Peer Review file · Nature Communications]

Bioadhesive and Conformable Bioelectronic Interfaces for Vasomotoricity Monitoring and Regulation

Corresponding Author: Professor Zhitao Zhou

Version 0:

Reviewer comments:

Reviewer #2

(Remarks to the Author)

Comments:

Wang et al. show an interesting study on the placement of a bioelectronic device with a bioadhesive for monitoring vascular electrophysiology in rabbits. These devices are constructed with PEDOT:PSS-coated electrodes on a poly(imide) backing with an attached poly(urethane)/silk fibroin adhesive layer. The mechanical properties of these devices were analyzed before in vivo testing. For in vivo testing, these devices were initially placed into contact with the abdominal aorta. This artery was subjected to a series of chemical stimuli to initiate an electrophysiological response, followed by an inter-ganglion transection, performed to reduce CNS-derived input to the artery. Following these acute tests, the devices were placed distal to an intra-arterial stent, and the electrophysiological response of the tissue was monitored over time after stent placement. Finally, the performed a closed-loop monitoring-stimulus study in the vicinity of a placed stent to augment vascular electrophysiology in this context. While this study potentially very interesting, I have a number of comments and questions about nature of the data and its repeatability before this manuscript could be considered suitable for publication.

1. In Fig. 1b, the schematic image of the device indicates that the adhesive layer is on the back of the device, i.e. away from the actual interface with vasculature. Is this the case for the actual device? If not, please change the schematic to show the role of the adhesive in these surgeries.
 - a. If so, what role does the adhesive play if it is in contact with the tissue surrounding the artery rather than providing a stable mechanical interface with the artery?
 - b. If the adhesive is not used to create an interface with the artery, please comment on the mechanical stability of the device for long-term recordings. Does the acquired signal result from mechanical artifacts, or is this an electrophysiological response?
2. The authors state that they measure an interfacial toughness of 21 and 139 N/m. The measurement performed (ASTM F2256) does not provide material toughness and instead provides a measure of interfacial strength. Please provide details on the conversion from this measurement to the data plotted in Fig. 1c.
3. Figs. 2, 3, and 4 all relate to in vivo testing data. No numbers are provided for the number of experiments or number of animals utilized for these tests. Please provide clear detail of your n for each test and calculation.
4. Figs. 2, 3, and 4 involve the in vivo recording of electrophysiological data derived from the adventitial side of the artery. What is the physiological origin of data collected in this fashion? For example, in Fig. 2b, the basal trace shows a series of ~2-3 peaks per second with an amplitude of ~0.2 mV. Is this heart rate (approximately the correct frequency)? If so, why is the heart amplitude so variable? The basal conditions shown in Fig. 2b, f, and j are substantially different. If this effect arises from the pulsatile nature of blood flow through the artery, then is it mechanical or electrophysiological? For example, why does a vasoconstrictive event like the application of norepinephrine, result in a larger signal amplitude?
5. In Fig. 3h, the authors display the relative power. How is this metric calculated? Similarly, Fig. 2h also has a relative amplitude. Relative to what? In general, this manuscript provides little detail or references for how the metrics are calculated. Please provide further detail on these points in the methods section.
6. Fig. 3h indicates that norepinephrine was delivered during the chronic experiments detailed in this figure. The methods make no mention of how this portion of the study was conducted. What was the purpose of this experiment, and what were the results? Please add detail to your methods indicating the experiments you performed. The methods should provide sufficient detail for readers to recreate your studies.

7. Fig. 4 details the closed-loop monitoring and stimulation of vascular electrophysiological activity. The authors monitor and then apply a stimulus to alter the monitored signal. In these cases, the authors make numerous references to a "healthy" signal. What constitutes a healthy signal, and how was this determined? Please provide detail on the expected and actual outputs for this study.

8. The authors also identify the 300 μ A stimulus as the most appropriate stimulus. Why? Further, why heart racing present as a lack of signal in the electrophysiological trace from the 500 μ A stimulus? Please provide further detail on what you are recording.

Reviewer #3

(Remarks to the Author)

Overview

The regulation of vasomotor function is essential for maintaining tissue homeostasis. Current clinical modalities/hemodynamic sensors have limited capture ability and are unable to define mechanisms of vasomotor dysfunction. In this regard, the authors report a bioadhesive-bioelectronic interface for high-fidelity monitoring and modulation of vasomotor activity. This interface is conformal, adheres stably to arteries and enables the precise recording of vasomotor states across a range of conditions (e.g. automatic nerve stimulation). To demonstrate clinical utility, the authors implement a stent implantation model to illustrate how their new interface effectively monitors/identifies abnormal vasomotor function. The authors conclude that their technology provides new insight on vasomotor function (in vivo) and could be used to advanced vascular disease management.

General Comments

The paper's premise is intriguing that being direct long-term monitoring of arterial vasomotor function in vivo. The observations are clear, the perturbations are straightforward and the conclusions are sound. The methods are meticulous and carefully describe, and in this regard the authors should be commended.

This manuscript has many positives but with one strategic weakness, that being the breath of applications to which the new technology can be applied. In this regard, the reviewer suggests significant expansion in two areas.

1) Inclusion of a second clinical scenario beyond the current stent model. One could consider an aortic dissection model, where the conduit vessel undergoes extensive matrix remodeling and expansion. This remodeling phenomenon can be induced by implanting mini-pumps and encoding them to release angiotension II so to induce hypertension.

2) Application of this technology to smaller resistance arteries or perhaps the venous circulation. In essence, how far can this technology be pushed and yet provide valuable physiological information.

Reviewer #4

(Remarks to the Author)

Overall, very interesting work on an implantable, bioelectronic interface to monitor the bioelectrical activity of vasomotor functions. The manuscript is generally clear and well-written. I recommend a minor revision to address the below comments:

1. The addition of PU to form the SF/PU composite decreased the young's modulus of SF. The authors state that a 10/6 ratio was used to achieve a young's modulus of <3 MPa to ensure mechanical compatibility with the aorta. Based on images of the device (Fig. 1d and Fig. S1) and the exploded-view in Fig. 1b, the overall device uses a solid thin film of PI. Thus the device likely has a young's modulus near 916 GPa similar to that measured for PI (Fig. S4).

a. Does the PI film dictate the modulus of the device?

b. If so, it would be helpful to measure the modulus of the entire device instead of only individual layers.

c. Is the silk substrate larger than the PI film and electrode layers? Is that what improves the adhesion to the 3 mm diameter tube? Since the interface consists of PI and PEDOT:PSS with direct contact to the tube/artery.

d. If the device does have a modulus larger than an artery/aorta, does wrapping the device around an artery constrict the expansion of the artery? If it's wrapped too loose, the electrical interface likely degrades.

2. Is there a reason 64 channels were used?

3. For Fig. S14, was the same device moved during the in vivo experiment or were three devices implanted?

4. Can it be confirmed that the changes in signal amplitude during the different studies (pharmacologic and stent) is not due to mechanical noise disturbing the electrodes?

5. Regarding the therapeutic effects of stimulation:

a. Was the stimulation only applied on day 14? More details on the stimulation timing/schedule are needed.

b. The manuscript indicates it is 'closed-loop', so is the stimulation automated based on the recorded signals? If so, as shown in Fig. 4a, what was considered 'dysfunction detection'?

c. In Fig 4g, how much time was there between the stimulation and the 'after' ultrasound image?

d. Is the enhancement of electrical signal conduction and reduction of arterial stiffness a permanent effect? Or does it dissipate with time? I'm not sure this requires additional animal studies, but comments should be added to the manuscript to clarify how long this effect was observed for after stimulation (minutes, hours, days, etc). (Side note: Future work on long-term stimulation and therapeutic effects would be interesting!)

Reviewer #5

(Remarks to the Author)

The paper by Xiner Wang et al., entitled "Bioadhesive and Conformable Bioelectronic Interfaces for Vasomotoricity Monitoring and Regulation," addresses the feasibility of using an implantable sensor made with PEDOT:PSS/PI/Electrodes/PI/Silk Sub. for vasomotoricity monitoring and stimulation.

Overall, the sensor design closely resembles existing structures, making its novelty somewhat limited. The study primarily presents preliminary findings on the sensor's potential for vasomotoricity monitoring and stimulation. However, in terms of technological maturity and practical usefulness, additional developments are necessary. These could include, for example, integrating wireless data transmission and power delivery, or identifying new findings and applications for the sensor.

To improve the quality of the paper, it is crucial to emphasize the unique advantages of the sensor in both sensing and stimulation, especially when compared to existing invasive methods. Additionally, in terms of application, new findings or innovative applications should be included.

Here are some comments that I think need to be properly addressed.

- 1) 33-35: Herein, we report a bioadhesive and conformable bioelectronic (BACE) interface with the incorporation of bioadhesive silk fibroin materials for high-fidelity monitoring and modulation of vasomotoricity.
> A supporting statement is needed that includes a clear rationale and supporting data to justify why the sensor enables high-fidelity monitoring. It is important to explain with specific reasons and provide relevant data or figures that demonstrate the extent to which high-performance monitoring is achievable with this sensor.
- 2) 36-38: The interface's conformal and stable adhesion to arteries in aqueous environments, combined with its exceptional electrical performance, enables precise recording of vasomotor states across varying conditions.
> Supporting details are needed regarding the exceptional electrical performance of the sensor. Please provide a clear rationale and be specific with the numerical data that demonstrate its outstanding electrical characteristics.
- 3) 38-40: In a stent implantation model, the BACE interface effectively monitors and identifies alterations in bioelectrical activity associated with abnormal vasomotor function, validated against clinical gold standards.
> Revise the sentence to make it clearer by providing specific examples (eg: clinical gold standards = arterial stiffness parameters in Ultrasound)
- 4) 42-44: These findings offer new insights into vasomotor electrophysiology and present a promising approach for advancing vascular disease management in biomedicine.
> Provide specific examples and clarify the details.
- 5) 71-74: While these methods provide detailed anatomical information crucial for diagnosing and treating vascular diseases, they offer limited insight into the dynamic regulation of vasomotor function under various physiological or pathological conditions.
> Provide more detailed explanations to clarify the reasons behind your statements. (why it provides limited insight?)
- 6) 79-91: Innovations in mechanical sensing technologies, including piezoresistive¹⁸, capacitive^{19,20}, piezoelectric^{21,22}, and triboelectric²³ sensors, have facilitated the measurement of vascular mechanical properties such as blood pressure, flow velocity, and pulse wave velocity. However, as a process governed by autonomic nerve activity, the vasomotor function relies fundamentally on electrophysiological events, which serve as the basis for neural signal transmission and the execution of physiological responses. Recent advances in implantable bioelectronic interfaces address these limitations by establishing direct contact with target tissues, enabling more precise measurement of physiological parameters compared to traditional methods. Despite these advancements, existing technologies are insufficient for directly and reliably recording high-quality vascular electrophysiological activity, which is essential for unraveling the electrophysiological mechanisms underlying the ANS regulation of vasomotor tone.
> Piezoresistive, capacitive, piezoelectric, and triboelectric sensors can also measure electrophysiological signals using different modalities. These sensors can directly and reliably record the signals, and even references 18, 20, and 21 allow wireless connectivity, providing convenient measurements.

> Please clarify your argument regarding the "recording of high-quality vascular electrophysiological activity." Why do you assert that your sensor provides high-quality recordings compared to the conventional sensors mentioned? If the sensing principles behind each technology are explained, along with a quantitative comparison of their pros and cons, and if the unique features of your developed sensor are highlighted with specific data or numbers, your argument will be much clearer.
- 7) 100-106: This design allows for intimate and stable wrapping around blood vessels in aqueous environments, facilitating reliable, high-quality acquisition of vascular electrophysiological (VE) signals.
> Additional explanation is required to support this argument. What technologies enable this high-performance capability, and to what extent does the sensor achieve high performance? Specific numerical data is needed to demonstrate the level of performance.

With its high-fidelity and high-resolution recording capabilities, the BACE interface enables precise monitoring of vasomotor function in distal arterial segments following the implantation of the metal stent with performance validated against the stiffness parameter, which is the clinical gold standard in arterial assessment.

> Evidence is needed to support the claim of high fidelity. In terms of high resolution, data is needed to show how the 64 implemented channels affect the results when compared to using fewer or more than 64 channels.

8) 109-111: Our approach holds significant potential for elucidating the electrophysiological mechanisms underlying vasomotor function and advancing both the clinical diagnosis and therapeutic intervention of vascular diseases.

9) 124-126: This innovation not only deepens the understanding of the bioelectrical mechanisms underlying vasomotor dynamics but also introduces promising therapeutic strategies for addressing vasomotor dysfunction.

> If specific applications and supporting evidence are added to the above statements, the overall quality of the paper would be significantly improved.

10) 114-126: Overview of the BACE interface Section

> If the electrophysiological signal sensing principles of the developed sensor, along with the mechanism of dysfunction remission through electrical stimulation, are explained, it would be easier for beginners to understand.

11) 192-194: Notably, the ratio of rising edge duration to falling edge duration within a single signal cycle exhibited characteristic patterns dependent on the vasomotion state.

> Does all the observed data reflect the same phenomenon? It would be helpful to explain how many data points were collected and include an analysis of how the trend changes with varying doses. Additionally, if possible, it would be beneficial to discuss the physiological significance of these findings.

12) 204-207: Denervation leads to a significant reduction in VE activity, while compensatory mechanisms involving intact ganglia and circuits help maintain partial arterial function, allowing for the detection of weak electrical signals (Fig. 2j and Fig. S10).

> In this case, it would be helpful to include whether electrical stimulation leads to an increase in the damaged vascular electrophysiological (VE) activity, as this would enhance the novelty of the paper.

13) Fig2.f : The basal condition appears small when compared to Fig. 2b and 2j. Is there a reason for this?

14) 227-232: The BACE interface conformed seamlessly to the aorta, ensuring intimate contact and reducing tissue-electrode impedance, with all 64 recording sites demonstrating impedance values below 10 k Ω (Fig. 3b).

> Is a contact impedance of 10k Ω considered good? Why or why not? What is the impedance level for other technologies? Including references or data would help clarify the explanation.

This exceptional electrical performance enabled reliable acquisition of vascular electrical activity associated with vasomotor states, yielding recordings characterized by low baseline noise and stable signal amplitudes (Fig. 3c).

> In terms of baseline noise and signal amplitudes, how do other technologies compare?

15) 238-240: In contrast, our multi-channel BACE interface enabled high-resolution tracking of electrophysiological changes in the distal aorta, directly reflecting vasomotor status.

> What are the effects of using multiple channels? An explanation of the correlation between the number of channels and their characteristics is needed, along with a clarification of what high-resolution tracking means. Additionally, data and further explanation should be provided to support this.

16) 295-298: These results suggest that electrical stimulation may enhance electrical signal conduction in dysfunctional vessels, thereby improving vascular elasticity and compliance. This presents a promising therapeutic strategy for mitigating vasomotor dysfunction.

> This appears to be a good outcome

Version 1:

Reviewer comments:

Reviewer #2

(Remarks to the Author)

The authors have mostly addressed my comments. With my improved understanding from the authors' response, I have two remaining comments relating to Fig. 4. 1) Interpretation of the "states" elicited from the subsequent input stimuli. Why does a neuromuscular stimulation result in a sustained relaxation(?) after the input stimulus? I do recognize the apparent improvement in flow, as evidenced in Fig 4g, but I would anticipate that the signal would go back to baseline after some time

for a neuromuscular stimulation. Instead, it appears that each stimulus application results in concurrent increases in the electrophysiological response that is sustained for some time past the authors recording. If this is a stimulation of the local junctions with vascular smooth muscle, why does the muscle not constrict(?) again after the stimulation period ends? Please provide some interpretation of these data and the sustained changes in the text. 2) Closed-loop system. In Fig. 4a, if the applied stimulus is not automated, as indicated in the authors' response, then calling this system "closed-loop" is somewhat of an exaggeration. I do not think this is consequential for the novelty of the manuscript, as the data collection suggests that a closed-loop system could be constructed, but the authors should adjust their wording to clarify.

Reviewer #3

(Remarks to the Author)

The authors provided an authoritative and clear response to my concerns.

Reviewer #4

(Remarks to the Author)

Looks good, recommend publishing

Reviewer #5

(Remarks to the Author)

Since spatial resolution appears to be a critical system specification, it would strengthen the manuscript if the authors could also discuss the potential effects of varying spatial resolutions. Currently the revised version only compares data at 4,8,16 channels under the same spatial resolutions.

Version 2:

Reviewer comments:

Reviewer #2

(Remarks to the Author)

The authors have addressed all of my concerns.

Point by point response (comments in black and responses in blue):

Reviewer #2:

[1] Wang et al. show an interesting study on the placement of a bioelectronic device with a bioadhesive for monitoring vascular electrophysiology in rabbits. These devices are constructed with PEDOT:PSS-coated electrodes on a poly(imide) backing with an attached poly(urethane)/silk fibroin adhesive layer. The mechanical properties of these devices were analyzed before in vivo testing. For in vivo testing, these devices were initially placed into contact with the abdominal aorta. This artery was subjected to a series of chemical stimuli to initiate an electrophysiological response, followed by an inter-ganglion transection, performed to reduce CNS-derived input to the artery. Following these acute tests, the devices were placed distal to an intra-arterial stent, and the electrophysiological response of the tissue was monitored over time after stent placement. Finally, the performed a closed-loop monitoring-stimulus study in the vicinity of a placed stent to augment vascular electrophysiology in this context. While this study potentially very interesting, I have a number of comments and questions about nature of the data and its repeatability before this manuscript could be considered suitable for publication.

We sincerely thank the reviewer for the positive comments. Regarding the concern about the nature of the data and its repeatability, we have included more comprehensive details in the point-by-point response below and in the revised manuscript.

[2] In Fig. 1b, the schematic image of the device indicates that the adhesive layer is on the back of the device, i.e. away from the actual interface with vasculature. Is this the case for the actual device? If not, please change the schematic to show the role of the adhesive in these surgeries.

a. If so, what role does the adhesive play if it is in contact with the tissue surrounding the artery rather than providing a stable mechanical interface with the artery?

We sincerely thank the reviewer for the insightful comments and we are sorry for any confusion and unclarity regarding the structure of the device and the role of the adhesive layer. First, we would like to clarify that the schematic in Fig. 1b is accurate in representing the placement of the adhesive layer on the back of the device. This adhesive layer is slightly larger than the outer contour of the polyimide and functions as an adhesive overcoat. Its outer edge adheres to the arterial wall, securely anchoring the device around the artery and ensure its stability during the procedure. Furthermore, given that the electrophysiological sensing principle relies on electrodes being in direct contact with biological tissues to record subtle biopotential changes, placing the adhesive layer on the inner side of the device, in direct contact with the artery, would disrupt the contact between the conductive layer and the tissue. Such interference could potentially compromise the functionality of the device. We have also revised the manuscript to more clearly describe the device structure and the role of the adhesive layer.

b. If the adhesive is not used to create an interface with the artery, please comment on the mechanical stability of the device for long-term recordings. Does the acquired signal result from mechanical artifacts, or is this an electrophysiological response?

We sincerely thank the reviewer for the valuable comments. As previously discussed, the silk fibroin adhesive layer plays an essential role in establishing a stable interface with the artery. It functions similarly to an overcoat, where the adhesive outer edge makes contact with the vessel wall, thereby

promoting stable mechanical coupling between the device and the artery. As demonstrated in **Figure 1d** and **Supplementary Figure 6** in revised manuscript, our *in vitro* tests indicate that devices with the silk fibroin layer maintain stable attachment around the cylindrical surface in liquid environments for up to two months, in contrast to devices without the adhesive layer. Thus, this long-term adhesion ensures that the mechanical stability does not pose a concern during extended recordings.

We sincerely appreciate the reviewer’s concern regarding the nature of the acquired signal, as we recognize that mechanical artifacts can negatively affect signal detection and interpretation in bioelectronic measurements. We would like to demonstrate that the acquired signal here is not mechanical artifacts but rather genuine electrophysiological responses through the following three points. Firstly, a low interfacial impedance can serve as an indicator of the conformability between the BACE interface and the vascular wall. Such intimate contact contributes to less relative mechanical sliding between the device and the artery, which is essential for accurate and reliable recording of electrophysiological signals with minimal contamination from motion artifacts [Zhao, Y. et al. *Nat. Commun.*, 12(1), 4880 (2021); Song, D. et al. *Adv. Mater.*, 35(48), 2304956 (2023); Wei, B. et al. *Cell Rep. Phys. Sci.*, 4(4), 101335 (2023)]. To validate the mechanical stability and integrity of the interface, we analyzed the interfacial impedance measured at 1 kHz during three trials conducted across the progression of the experiment in multiple rabbits. As shown in **Figure R1**, the impedance values remained consistently low and exhibited no statistically significant variation across trials, indicating that the interface maintained reliable conformal contact and stable electrical coupling with the artery throughout the experimental process without substantial motion-induced disruption.

Figure R1. Stability of interfacial impedance as the experiments progress. The interfacial impedance measured at 1 kHz during the three trials throughout the course of the experiment across $n = 3$ rabbits. Data are presented as box plots which depict the data median (center line), upper and lower quartiles (box bounds), 1.5 times the interquartile range (whiskers) and outlier values beyond this range (circles). p values for impedance comparison: $p = 0.4508$ (Trial 1 vs. Trials 2), $p = 0.2154$ (Trial 1 vs. Trial 3), $p = 0.1258$ (Trial 2 vs. Trial 3). (ns, $p > 0.05$, * $p \leq 0.05$, ** $p \leq 0.01$, *** $p \leq 0.001$, and **** $p \leq 0.0001$, Wilcoxon signed-rank test)

Secondly, interference sources stemming from physiological functions, such as heartbeat and respiration, typically exhibit small amplitudes and occur within specific frequency ranges, thus

coexisting with electrophysiological signals across different frequency bands [Yin, J. et al. *Nat. Rev. Bioeng.*, 2(7), 541-558 (2024); Park, B. et al. *Science*, 376(6593), 624-629 (2022)]. Thus, the signals can be post-processed employing filtering algorithms to separate the contributions of these overlapping functions and provide valuable physiological information [Tian, L. et al. *Nat. Biomed. Eng.*, 3(3), 194-205 (2019)]. In our study, Butterworth-based band-pass filters equipped with cutoff frequencies that cover the target signals can minimize undesired frequency components associated with artifacts. Meanwhile, due to the global synchronization of artifacts induced by factors such as heartbeats, common-mode noise caused by mechanical interference can be effectively reduced using either common average reference (CAR) or common median reference (CMR) [Zhao, Z. et al. *Nat. Biomed. Eng.*, 7(4), 520-532 (2023); Xu, Y. et al. *Nat. Biomed. Eng.*, 7(10), 1307-1320 (2023)]. As discussed in the Methods section, we utilized CAR to reduce common-mode noise in multi-channel data.

Lastly, further evidence supporting that the acquired signals are not mechanical artifacts can be found in our experimental results. As illustrated in **Figure 2i** and **2j**, transection of the sympathetic ganglion did not induce immediate or substantial changes in macroscopic physiological parameters such as heart rate or blood pressure. However, a marked attenuation was observed in the recorded bioelectrical signals following the intervention, indicating the disruption in the electrical activity of the sympathetic nervous system despite the relative stability of hemodynamics. In addition, as shown in **Figure 4c**, electrical stimulation at an amplitude of 500 μA elicited observable physiological responses including accelerated heart rate and deepened respiration. These pronounced cardiorespiratory changes were not accompanied by corresponding increase in the amplitude or frequency of the recorded signals, suggesting that the acquired signals are electrophysiological in nature rather than being attributed to mechanical disturbances. Thank again for the reviewer to help us clarify this important aspect of our work.

[3] The authors state that they measure an interfacial toughness of 21 and 139 N/m. The measurement performed (ASTM F2256) does not provide material toughness and instead provides a measure of interfacial strength. Please provide details on the conversion from this measurement to the data plotted in Fig. 1c.

We thank the reviewer for the valuable comment regarding the interpretation of our adhesive measurements and the terminology used to describe the interfacial properties. Material toughness generally refers to the ability of a bulk material to absorb energy and undergo plastic deformation before fracture. It is typically quantified as the area under the stress-strain curve and expressed in units of energy per unit volume (e.g., J/m^3). Interfacial toughness and interfacial strength are both critical parameters for characterizing adhesion performance, although they capture slightly different aspects of interfacial behavior. Interfacial toughness describes the energy required to propagate a crack along the interface between two bonded materials and is quantified as energy per unit area (e.g., J/m^2 or N/m) [Wang, C. et al. *Science*, 377(6605), 517-523 (2022); Li, J. et al. *Science*, 357(6349), 378-381 (2017)]. On the other hand, interfacial strength refers to the maximum stress or force per unit area the interface can withstand before the onset of mechanical failure.

In our study, interfacial toughness was determined using a peeling test, which is well-established for evaluating the energy release rate during crack propagation. In such tests, as the peeling process reaches a steady state, the measured force typically stabilized at a plateau with minor

oscillations. For 180° peel test configuration, the interfacial toughness is calculated by dividing twice the plateau force by the width of the sample, as described in previous studies [Yuk, H. et al. *Nature*, 575(7781), 169-174 (2019); Deng, J. et al. *Nat. Mater.*, 20(2), 229-236 (2021); Wu, J. et al. *Sci. Transl. Med.*, 14(630), eabh2857 (2022)].

Accordingly, the values of 21 N/m and 139 N/m reported in **Figure 1c** were derived from the measured steady-state peel forces following this established methodology, as detailed in the Methods section and illustrated in **Supplementary Figure 2**. The use of the term “interfacial toughness” in this context is consistent with the conventions adopted in the aforementioned literatures involving T-peel tests for flexible films or biointerfaces.

[4] Figs. 2, 3, and 4 all relate to *in vivo* testing data. No numbers are provided for the number of experiments or number of animals utilized for these tests. Please provide clear detail of your *n* for each test and calculation.

We sincerely thank the reviewer for the insightful comment and we are sorry for the lack of clarity regarding the number of experiments or animals used in our original manuscript. We fully recognize that transparent reporting of sample size is essential for assessing the reliability and reproducibility of our *in vivo* findings. To ensure the robustness of our data, we conducted *in vivo* experiments using a minimum of three rabbits ($n \geq 3$). The time-domain waveforms and time-frequency spectrograms presented in both the main figures and supplementary information display the representative electrophysiological responses observed in the experiments. For the statistical analyses presented in Figs. 2, 3, and 4, the corresponding number of animals has now been explicitly clarified in the main texts and figure legends of the revised manuscript. Additionally, a detailed description of the sample sizes and the corresponding representative data for animals not depicted in the main Figs. 2, 3, and 4 are provided as follows.

In Figure 2, the experiments were performed on three rabbits ($n = 3$) to monitor vasomotor dynamics under various intervention protocols. Specifically, **Figures 2d** and **2h** in the revised manuscript analyze the ratio of vascular electrophysiological (VE) signal power before and after the administration of different doses of norepinephrine (NE) and acetylcholine (ACh) (i.e., 0.1, 0.2, and 0.4 mL) in three rabbits, revealing a consistent dose-dependent effect on both vasoconstriction and vasodilatation across individuals. The revised **Figures 2b-c** and **2f-g** illustrate the typical time-domain and time-frequency responses of VE signals obtained from one of the three rabbits under varying doses of NE and ACh. To ensure comprehensive reporting, the corresponding results from the other two rabbits under the same experimental conditions are shown in **Figure R2**. With increasing doses of these vasoactive agents, both vascular contraction and dilation are enhanced, as evidenced by the progressively larger amplitudes and higher low-frequency power of the VE signals recorded via the BACE interface. On the other hand, marked differences can be observed from the enlarged view of the signals under different pharmacological influences. Likewise, **Figure R3** displays the time-domain and time-frequency characteristics of the VE signals for the remaining two rabbits before and after sympathetic ganglionectomy, corresponding to **Figure 2j**. Following the severing of the sympathetic nerve pathways, a noticeable reduction in the amplitude and low-frequency power of the signals was observed, indicating a diminished vasoconstrictive response, which aligns with the known physiological effects of sympathetic nervous system disruption. To further assess the capability of the BACE interface in accurately monitoring vasomotor activity, we also applied t-distributed stochastic neighbor embedding (t-SNE) to the feature sets extracted from

the remaining rabbits under varying degrees of vasoconstriction and vasodilatation, similar to the analysis presented in **Figure 2k**. The results in **Figure R4** validate the BACE interface's sensitivity in distinguishing subtle differences in vasomotor responses induced by pharmacological and surgical interventions. These findings were consistently observed across all three rabbits, further supporting the reproducibility and reliability of the electrophysiological recording via the BACE interface.

Figure R2. Vascular electrical responses evoked by varying doses of different vasoactive agents. The representative time-domain waveforms and time-frequency spectrograms of the VE signals recorded under basal condition and after NE and ACh delivery from two rabbits (left) highlight the dose-dependent effects on vasomotor function. The zoomed-in view of the time-domain waveforms (right) provides a more detailed comparison of the amplitude and rhythm of the responses induced by different drugs at different doses.

Figure R3. Vascular electrical responses before and after sympathetic ganglionectomy. The representative time-domain waveforms and corresponding time-frequency spectrograms before and after severing the sympathetic path ways from two rabbits, indicative of the vasomotor function alteration.

Figure R4. Two-dimensional t-SNE feature distribution of the VE signals under pharmacological and surgical interventions. The plots illustrate the clustering of seven distinct electrophysiological features induced by varying doses of NE and ACh, as well as the effects of sympathetic ganglion transection, showing the separation between different vasomotor states of constriction and dilation.

In Figure 3, the effect of stent implantation on distal arterial vasomotor function was assessed in three rabbits ($n = 3$). **Figure 3g** in the revised manuscript displays the normalized power of the VE signals recorded from the distal arteries of all three rabbits before, immediately after, and one

month following stent implantation. These results were subjected to statistical analysis, which revealed a consistent trend across individuals. Specifically, the vascular electrical activity gradually diminished, reflecting a progressive deterioration in vasomotor function over time. In alignment with this trend, the representative VE responses from all three rabbits across the different time points are presented in the main figures (**Figures 3d-f**) and supplementary information (**Figure S16**). The consistency of these findings across multiple subjects enhances the robustness of the data and further demonstrates the reliability of the BACE interface in monitoring and quantifying changes in vasomotoricity following stent implantation.

In Figure 4, for the closed-loop regulation of vascular dysfunction, three rabbits ($n = 3$) were used to evaluate the effect of electrical stimulation on distal arteries following stent implantation. The time-domain signals of the VE responses to electrical stimulation are shown in the main figures (**Figures 4d-e**) and **Figure R5**, which provide a detailed view of the VE signal dynamics before and after stimulation. To further assess the effect of repeated electrical stimulation, we extracted power features from the VE signals of all three rabbits under various states (State 1 - State 5), corresponding to different stages of electrical stimulation. These features were then normalized and subjected to statistical analysis, with the resulting bar chart presented in the revised **Figure 4f**. While individual variations in the vascular responses to each stimulation were observed, the overall trend indicated that electrical stimulation was effective in promoting the recovery of vascular electrical activity in the distal arteries. These findings underscore the potential of electrical stimulation as a promising approach to restoring vasomotor dysfunction following vascular intervention.

Figure R5. Electrical regulation of vasomotor dysfunction. The VE signals recorded by the BACE interface from the distal segment of the artery in response to a series of electrical stimulations with a charge-balanced current pulse (pulse amplitude: $300 \mu\text{A}$, pulse width: $100 \mu\text{s}$) conducted 14 days after stent implantation from two rabbits. Red stars mark the timing of the stimulation events.

We hope these revisions could provide a clear understanding of the experimental protocols including the sample sizes used for each *in vivo* testing. We appreciate the reviewer's valuable suggestion to help us to strengthen the rigor of the manuscript.

[5] Figs. 2, 3, and 4 involve the *in vivo* recording of electrophysiological data derived from the adventitial side of the artery. What is the physiological origin of data collected in this fashion? For example, in Fig. 2b, the basal trace shows a series of ~2-3 peaks per second with an amplitude of ~0.2 mV. Is this heart rate (approximately the correct frequency)? If so, why is the heart amplitude so variable? The basal conditions shown in Fig. 2b, f, and j are substantially different. If this effect arises from the pulsatile nature of blood flow through the artery, then is it mechanical or electrophysiological? For example, why does a vasoconstrictive event like the application of norepinephrine, result in a larger signal amplitude?

We sincerely thank the reviewer for the valuable comments. Firstly, regarding the physiological origin of the data in our study, the adventitia (outer wall of a blood vessel) contains postganglionic sympathetic nerves that travel along the arteries and play a critical role in regulating both vasoconstriction and vasodilation [Thomas, G. D. *Adv. Physiol. Educ.*, 35(1), 28-32 (2011)]. The dynamic regulation of vascular tone, which refers to the contraction and relaxation of vascular smooth muscle cells within the walls of arteries, is governed primarily by the balance between the sympathetic and parasympathetic branches of the autonomic nervous system [Finsterer, J. *Comparative Medicine: Anatomy and Physiology*, 45-60 (2013); Sheng, Y. et al. *Int. J. Physiol. Pathophysiol. Pharmacol.*, 10(1), 17 (2018); Jackson, W. F. *Hypertension*, 35(1), 173-178 (2000)]. Hence, the electrical activity recorded in our study arises from the sympathetic nerve fibers in the adventitia, reflecting their autonomic regulation of the smooth muscle through neurotransmitter release and highlighting the integral role of sympathetic innervation in controlling vasomotoricity.

[Figure Redaction]

Figure R6. Schematic diagram of vasoconstriction and vasodilation controlled by sympathetic nerve fibers [Finsterer, J. *Comparative Medicine: Anatomy and Physiology*, 45-60 (2013)].

Secondly, while the basal trace shown in **Figure 2b** of the original manuscript does indeed show a series of peaks in the range of 2-3 peaks per second which resembles the frequency of heart rate, these peaks are not directly indicative of the heart rate itself. Instead, the observed frequency likely reflects electrophysiological responses related to the sympathetic nerve activity that regulates vascular tone. The periodicity of the signal may be attributed to the synchronization of nerve firing with the rhythmic pulsations of blood flow, as the sympathetic nervous system modulates vascular tone in response to changes in cardiovascular demands. In addition to the above interpretation, we also employed a filtering algorithm to further validate our results. Specifically, we applied a band-pass filter to remove frequency components below 5 Hz, which includes the 2-3 Hz frequency range observed in the basal trace. As shown in **Figure R7**, the filtered signal (after removing components below 5 Hz) retains the same general amplitude fluctuations as the original signal, indicating that the filtering process does not alter the primary trends observed in the data. Consequently, this analysis confirms that the presence of low-frequency components, which resemble the frequency range of heart rate, does not significantly influence the amplitude variation and affect the conclusions drawn from our experiments.

Figure R7. Comparison of the original and filtered signals. **a**, Representative time-domain traces of the original signal (0.5-50 Hz) and the filtered signal (5-50 Hz) before and after NE delivery. **b**, Power spectral density (PSD) curves of the signals shown in **a** to verify the presence or absence of low-frequency components below 5 Hz.

Moreover, we would like to clarify that the variability in the signal amplitude is not a reflection of the heart rate amplitude fluctuations. Instead, it is more plausibly attributed to the activation and modulation of sympathetic nerve fibers, which dynamically regulate the contractile behavior of vascular smooth muscle. The differences in baseline conditions across **Figures 2b, f, and j** can be attributed to several factors, including inter-individual variations among the rabbits, as well as variances in the depth of anesthesia which can influence the autonomic nervous system responsiveness [**Bankenahally, R. et al. *Bja Education*, 16(11), 381-387 (2016)**]. While such baseline variability is expected in *in vivo* recordings, our analysis emphasizes the relative changes in signal amplitude before and after pharmacological or surgical interventions, which provide a more robust and physiologically meaningful basis for interpreting the results in our study.

Lastly, to further substantiate the origin of our data, we conducted an *in vitro* experiment where the BACE interface was wrapped around an elastic synthetic vessel and fluid was cyclically withdrawn and injected using a syringe to simulate the contraction and dilation of the blood vessel (**Figure R8a**). This process applied dynamic pressure to the vessel wall, thereby mimicking the mechanical effects of blood flow. The change in external diameter of the synthetic vessel (~ 0.3 mm) during simulated contraction and relaxation closely approximates the physiological range of systolic-diastolic diameter variation observed in the abdominal aorta of live rabbits [**Rahmani-Cherati, T. et al. *J. Tehran Heart Cent.*, 7(3), 128 (2012)**]. Importantly, measurements of interfacial impedance and baseline signals (**Figure R8b-d**) demonstrate remarkable stability under these pressure conditions, indicating that the BACE interface does not produce spurious signals in response to vessel wall deformation. These findings further support our interpretation that the recorded signals are electrophysiological in nature, arising from the electrical activity of perivascular sympathetic fibers rather than from mechanical artifacts induced by the blood flow pulsations. Therefore, the increase in signal amplitude following the application of norepinephrine is consistent with its known pharmacological action in mimicking sympathetic nerve stimulation. Norepinephrine significantly enhances vascular tone and leads to vasoconstriction by activating α -adrenergic receptors on the surface of vascular smooth muscle cells. This physiological response alters the local electrical environment, thereby modulating the recorded signal.

Figure R8. *In vitro* experiment simulating vasoconstriction and vasodilation. **a**, Experimental setup showing the BACE interface wrapped around a synthetic vessel. Water is withdrawn and injected to simulate vessel contraction and dilation. Red arrows indicate the direction of fluid movement. **b**, Representative electrophysiological traces recorded during a quiescent period and during cyclic fluid pumping. The baseline signal remained stable with minimal amplitude drift when the interface is subjected to fluid-induced pressure. **c**, Impedance measurements during the phase of quiescence and fluid pumping. **d**, Quantitative comparison of baseline signal RMS amplitudes under quiescent and pumping conditions. Data in **c** and **d** are presented as mean values \pm SD. p values for data in **c** and **d**: $p = 0.9234$ (impedance comparison) and $p = 0.4656$ (baseline RMS comparison). (ns, : $p > 0.05$, * $p < 0.05$, ** $p < 0.01$, *** $p < 0.001$, **** $p < 0.0001$, Wilcoxon signed-rank test)

We thank the reviewer again for the important comments which help us clarify the physiological origin of the recorded signals in our manuscript.

[6] In Fig. 3h, the authors display the relative power. How is this metric calculated? Similarly, Fig. 2h also has a relative amplitude. Relative to what? In general, this manuscript provides little detail or references for how the metrics are calculated. Please provide further detail on these points in the methods section.

We sincerely thank the reviewer for the valuable comments and would like to apologize for the insufficient methodological detail provided in the original manuscript regarding the calculation of the metrics presented in **Figure 2h** and **3h**. First, we would like to clarify that the metric shown in **Figure 3h** (relative power) and those shown in **Figure 2d** and **2h** were calculated using the same

approach. In the revised manuscript, we have corrected the y-axis labels in **Figure 2d** and **2h** to explicitly indicate that the values represent relative power, ensuring consistency across figures.

With regard to the definition and computation of relative power, the term "relative" refers to a within-subject comparison before and after a specific intervention or stimulation (e.g., pharmacological administration or surgical manipulation). For each subject, we identified two stable segments of signal-one preceding and one following the intervention. Within each segment, the signal's power was estimated by computing the power spectral density over the frequency range of 0.5-50 Hz using a sliding window approach with a window length of 100 ms and a 50% overlap. The power of each window was computed and then averaged across the entire segment to obtain a representative power value for both pre- and post-intervention periods. The relative power was defined as the ratio of the post-intervention average power to the pre-intervention average power for each subject. This approach yields a dimensionless quantity (unitless ratio), allowing us to normalize inter-subject variability and focus on the proportional changes in signal power induced by the experimental manipulation. We appreciate the opportunity to clarify these points and have now included the detailed description of the calculation into the Methods section of the revised manuscript.

[7] Fig. 3h indicates that norepinephrine was delivered during the chronic experiments detailed in this figure. The methods make no mention of how this portion of the study was conducted. What was the purpose of this experiment, and what were the results? Please add detail to your methods indicating the experiments you performed. The methods should provide sufficient detail for readers to recreate your studies.

We thank the reviewer for the insightful comments and we are sorry for the omission of methodological details regarding the experiment presented in **Figure 3h**. The delivery method for norepinephrine (NE) in this experiment was consistent with that used in the acute-phase studies described in **Figure 2a-d**. Specifically, NE was applied to the adventitial surface of the abdominal aorta to pharmacologically mimic the sympathetic neurotransmitter release. The principal aim of this experiment was to assess whether NE could modulate vascular electrophysiological activity following long-term stent implantation.

This investigation was motivated by our prior observation shown in **Figure 3g**, indicating that stent implantation compromises vascular autoregulation and dampens vasomotor responses mediated by the autonomic nervous system (ANS). In physiologically intact vessels, as demonstrated in **Figure 2a-d** and under normal condition in **Figure 3h**, NE typically induces significant electrophysiological changes associated with sympathetic-driven vasoconstriction. However, in stented vessels, we observed that NE failed to elicit comparable changes in signal power, suggesting a loss of functional responsiveness to sympathetic modulation. This finding was critical in informing the rationale for the subsequent development of a closed-loop electrical modulation strategy, which we propose as a potential therapeutic approach to restore vasomotor function in structurally compromised vessels.

In this experiment, 0.2 mL of norepinephrine was carefully applied to the aortic adventitia and the vascular electrophysiological signals were continuously recorded using the BACE interface before and after NE delivery. Signal processing followed the same procedure described in our previous response, wherein relative power was calculated by comparing the post-intervention signal energy to the pre-intervention baseline, providing a normalized metric of electrophysiological

change. These relative power values were then used to quantify the vascular response to NE under different physiological conditions. As shown in **Figure 3h**, we compared responses across three groups: healthy (non-stented) rabbits, rabbits immediately following stent implantation, and those one month post-implantation. The results reveal a marked decline in NE responsiveness in stented vessels, supporting the conclusion that stent implantation leads to sustained vasomotor dysfunction and raising a pressing need for alternative regulatory interventions. This methodological detail has now been added to the Methods section of the revised manuscript to enhance clarity and reproducibility.

[8] Fig. 4 details the closed-loop monitoring and stimulation of vascular electrophysiological activity. The authors monitor and then apply a stimulus to alter the monitored signal. In these cases, the authors make numerous references to a “healthy” signal. What constitutes a healthy signal, and how was this determined? Please provide detail on the expected and actual outputs for this study.

We sincerely thank the reviewer for the valuable comments and apologize for not providing sufficient detail regarding the definition and characterization of the “healthy” signal referenced in **Figure 4**. In the original manuscript (**Line 271**), the term “healthy state” refers to the baseline electrophysiological pattern recorded from the abdominal aorta of non-stented rabbits under basal physiological conditions. This signal profile is characterized by relatively high and stable amplitude, along with preserved responsiveness to sympathetic stimulation (e.g., norepinephrine application) as demonstrated in **Figure 3d** and **3h**. These signal features were consistently observed across multiple animals and served as the reference template for physiological vasomotor function. When real-time recordings from distal arterial segments deviated substantially from this reference—most notably through a decline in signal amplitude—feedback-controlled electrical stimulation was triggered to modulate the vasomotor state.

The expected output of the closed-loop regulation system was the recovery of an electrophysiological signal approximating the defined “healthy” state, characterized by restored amplitude and temporal stability. Our actual findings confirmed this expectation: following a limited number of stimulations, the monitored signals exhibited a progressive enhancement in amplitude, as shown in **Figure 4d-f**, indicating a measurable improvement in functional status. This electrophysiological recovery was further substantiated by ultrasound imaging and arterial stiffness assessment, which demonstrated a reduction in the stiffness parameter β , consistent with improved vascular compliance (**Figure 4g**). These outcomes demonstrate that our definition of a “healthy” signal is grounded in both empirical baseline recordings and functional recovery benchmarks, and that it served effectively as the control target for guiding closed-loop intervention. We have now included a more detailed explanation of the criteria and analysis used to define the healthy signal profile in the Results and Methods sections of the revised manuscript.

[9] The authors also identify the 300 μA stimulus as the most appropriate stimulus. Why? Further, why heart racing present as a lack of signal in the electrophysiological trace from the 500 μA stimulus? Please provide further detail on what you are recording.

We sincerely thank the reviewer for the thoughtful comments. To further substantiate the selection of 300 μA as the most appropriate stimulation parameter, we extended our original experiments to include a finer gradient of current amplitudes (50 μA) ranging from 100 to 500 μA . We systematically evaluated the distal vascular electrophysiological responses and assessed the effectiveness of each current level using both time-domain waveform observations and quantitative

analysis of signal power. As shown in **Figure R9**, stimulation at 300 μA elicited the relatively most pronounced enhancements in signal amplitude and power across multiple channels. Moreover, statistical analysis of the relative power before and after stimulation confirmed that 300 μA produced the highest median increase, outperforming all other tested current levels.

Figure R9. Optimization of stimulation current amplitude for restoring distal vascular electrophysiological activity in stented arteries. **a**, Representative time-domain traces of electrical signals recorded from the distal aorta of stented rabbits following electrical stimulation at different current amplitudes ranging from 100 μA to 500 μA , in 50 μA increments. **b**, Quantitative analysis of relative signal power (post- vs. pre-stimulation) at each stimulation amplitude. The red shaded region and dotted box highlight the selected optimal amplitude of 300 μA . Box plots in **b** depict the data median (center line), upper and lower quartiles (box bounds), 1.5 times the interquartile range (whiskers) and outlier values beyond this range (circles).

In addition to stimulation efficacy, we monitored physiological safety indicators, particularly heart rate. During 300 μA stimulation, heart rate remained stable at approximately 90 beats per minute, whereas 500 μA stimulation led to a significant increase to approximately 130 bpm. Despite the elevated heart rate under 500 μA , no corresponding increase in signal amplitude was observed. This dissociation between cardiac activity and electrophysiological signal output suggests that the

recorded signals are not artifacts of mechanical pulsation or cardiac motion, but rather reflect genuine electrophysiological activity, likely originating from postganglionic sympathetic nerve fibers located at the adventitial-medial border of the artery.

In summary, the selection of 300 μA was based on its ability to provide a trade-off between neuromodulatory efficacy and physiological safety, as higher current levels such as 500 μA introduced potential cardiovascular side effects without enhancing modulation efficacy. We have now included these additional results and a more detailed discussion of the signal's physiological origin in the revised manuscript. We appreciate the reviewer's insightful feedback which enables us to clarify and strengthen this key aspect of our study.

Reviewer #3:

Overview

[1] The regulation of vasomotor function is essential for maintaining tissue homeostasis. Current clinical modalities/hemodynamic sensors have limited capture ability and are unable to define mechanisms of vasomotor dysfunction. In this regard, the authors report a bioadhesive-bioelectronic interface for high-fidelity monitoring and modulation of vasomotor activity. This interface is conformal, adheres stably to arteries and enables the precise recording of vasomotor states across a range of conditions (e.g. automatic nerve stimulation). To demonstrate clinical utility, the authors implement a stent implantation model to illustrate how their new interface effectively monitors/identifies abnormal vasomotor function. The authors conclude that their technology provides new insight on vasomotor function(in vivo) and could be used to advanced vascular disease management.

We sincerely thank the reviewer for the insightful summary and comments on our work and we hope our findings could contribute to advancing the understanding and management of vasomotor dysfunction.

General Comments

[2] The paper's premise is intriguing that being direct long-term monitoring of arterial vasomotor function in vivo. The observations are clear, the perturbations are straightforward and the conclusions are sound. The methods are meticulous and carefully describe, and in this regard the authors should be commended.

We are truly grateful for the reviewer's positive and encouraging remarks on the manuscript.

[3] This manuscript has many positives but with one strategic weakness, that being the breath of applications to which the new technology can be applied. In this regard, the reviewer suggests significant expansion in two areas.

We sincerely thank the reviewer for the constructive comment. While the primary focus of our current study is on elucidating the electrophysiological mechanisms underlying vasomotor dysfunction and developing corresponding closed-loop modulation strategies, we fully acknowledge the reviewer's important observation regarding the breadth of potential applications. To address this, we have now expanded the revised manuscript to incorporate two additional disease models and physiological scenarios, which serve to demonstrate the scalability and adaptability of our bioelectronic interface. In parallel, we are actively developing wireless power and data transmission modules, with the long-term goal of achieving a fully implantable and wirelessly operated system, which will significantly broaden the translational potential of our platform. We are grateful for the reviewer's suggestions, which have guided us in strengthening the relevance and future outlook of our work.

[4] 1) Inclusion of a second clinical scenario beyond the current stent model. One could consider an aortic dissection model, where the conduit vessel undergoes extensive matrix remodeling and expansion. This remodeling phenomenon can be induced by implanting mini-pumps and encoding them to release angiotension II so to induce hypertension.

We thank the reviewer for the insightful suggestion regarding the inclusion of additional clinical models beyond the stent implantation scenario. We fully agree that incorporating disease models

involving matrix remodeling and vascular expansion—such as abdominal aortic dissection (AAD) and abdominal aortic aneurysm (AAA)—is essential for demonstrating the broader applicability of our bioelectronic interface.

We sincerely apologize to the reviewer for not being able to successfully establish the AAD model as initially intended. We did attempt to create AAD models, but due to certain technical challenges, we were unable to achieve the desired AAD model outcomes. Considering that AAD and AAA share overlapping pathological features such as extracellular matrix degradation and vessel wall expansion [Maguire, E. M. et al. *Pharmaceuticals*, 12(3), 118 (2019)], and given the extensive validation and widespread use of the AAA model within the vascular biology field, which provides a well-established framework for comparative analysis [Qian, W. et al. *Nat. Commun.*, 13(1), 512 (2022); Tian, Z. et al. *Cell Host & Microbe*, 30(10), 1450-1463 (2022)], we ultimately chose to focus our in-depth investigations on the AAA model. In our study, we induced AAA models in both mice and rabbits using distinct approaches. In mice, AAA was induced through oral administration of β -aminopropionitrile monofumarate (BAPN) at a dose of 1 g/kg per day, following established protocols [Pan, L. et al. *Circulation*, 145(9), 659-674 (2022)]. The BAPN treatment compromises structural proteins such as elastin and collagen, leading to medial degeneration and subsequent aneurysm formation. In rabbits, AAA was generated by topical application of papain (papaya-derived protease) directly onto the exposed abdominal aorta. Papain enzymatically degrades extracellular matrix components, promoting localized vessel wall weakening and aneurysmal dilation.

Figure R10. Monitoring and comparison of vasomotor activity in AAA-affected aortas using the BACE interface. **a**, Photograph of the BACE interface implanted on the abdominal aorta of a mouse with abdominal aortic aneurysm (AAA). **b**, Representative time-domain traces of electrophysiological signals recorded from the control (healthy) and AAA aortas. Quantitative analysis of root mean square (RMS) power (**c**) and frequency (**d**) obtained from the electrical signals recorded from the control (healthy) and AAA aortas. **e**, Photograph of the exposed AAA in a rabbit

model alongside its corresponding ultrasound image. Key anatomical landmarks including the proximal aorta, aneurysm region, and distal aorta are clearly annotated with measured vessel diameters. **f**, Representative time-domain vascular electrophysiological waveforms recorded from the proximal aorta, aneurysm site, and distal aorta using the BACE interface. Quantitative analysis of RMS power (**g**) and frequency (**h**) obtained from the electrical signals recorded from the proximal aorta, aneurysm site, and distal aorta. Data in **c** and **d** are presented as mean values \pm SD. Box plots in **g** and **h** depict the data median (center line), upper and lower quartiles (box bounds), 1.5 times the interquartile range (whiskers) and outlier values beyond this range (circles). p values for RMS power comparison: $p = 4.798 \times 10^{-5}$ (Proximal vs. AAA), $p = 2.678 \times 10^{-15}$ (Proximal vs. Distal), $p = 5.831 \times 10^{-4}$ (AAA vs. Distal). p values for RMS frequency comparison: $p = 0.0011$ (Proximal vs. AAA), $p = 3.223 \times 10^{-12}$ (Proximal vs. Distal), $p = 0.0011$ (AAA vs. Distal). (* $p < 0.05$, ** $p < 0.01$, *** $p < 0.001$, **** $p < 0.0001$, Friedman test with Dunn's post hoc test).

For both species, we designed our experiments to investigate vascular electrophysiological alterations from two complementary perspectives. In mice, we compared electrophysiological signals recorded from healthy aortas and BAPN-induced AAA models to characterize disease-associated changes (**Figure R10a**). As shown in **Figure R10b**, the electrophysiological signals recorded from healthy aortas exhibited clear rhythmic characteristics, indicative of normal vasomotor activity. In contrast, the signals from AAA aortas were notably flatter with a loss of key rhythmic features, suggesting impaired vasomotor function. To further characterize these differences, we analyzed both time-domain and frequency-domain features of the signals (**Figure R10c-d**). The root mean square (RMS) power in the time domain was higher in the healthy group while the frequency-domain RMS frequency (RMSF) was found to be lower in healthy arteries, indicating that AAA arteries exhibited less defined, more frequency-dispersed electrical activity.

In rabbits, we employed a self-controlled design, recording electrophysiological signals at three spatially distinct vascular segments: the proximal abdominal aorta, the aneurysm site, and the distal abdominal aorta (**Figure R10e**). This approach allowed us to examine regional heterogeneity within the same subject and assess how the presence of an aneurysm influences local vascular electrophysiology along the affected vessel. Notably, we observed a progressive decline in signal amplitude and RMS power from proximal to distal segments (**Figure R10f-g**). Particularly, the distal abdominal aorta displayed further diminished electrical activity, suggesting that the aneurysmal pathology may have a downstream effect on vascular electrophysiological conduction and smooth muscle responsiveness. Conversely, the RMS frequency exhibited an increasing trend moving distally, implying that the vascular electrical signals became more frequency-dispersed and less regular. This shift in frequency characteristics may reflect altered conduction properties within the vascular wall caused by structural remodeling and degeneration associated with the aneurysm.

These complementary models and analytical approaches provide robust evidence that our BACE interface sensitively detects nuanced electrophysiological changes associated with AAA pathology across species and spatial domains. These findings are likely due to the structural weakening, endothelial dysfunction and loss of vascular smooth muscle integrity in the aneurysmal aorta, which compromise the artery's ability to effectively contract and dilate in response to blood flow demands [Bailey, T. G. et al. *Am. J. Physiol. Heart Circ. Physiol.*, 314(1), H19-H30 (2018)].

We have incorporated this detailed description and analysis into the revised manuscript to demonstrate the versatility of our platform. However, to fully understand the underlying mechanisms of AAA progression, particularly in terms of matrix remodeling and vascular expansion,

additional experiments will be required. Our ongoing work will further elucidate these mechanisms and extend the use of the BACE interface to monitor vasomotor dysfunction in aortic dissection models as well.

[5] 2) Application of this technology to smaller resistance arteries or perhaps the venous circulation. In essence, how far can this technology be pushed and yet provide valuable physiological information.

We sincerely thank the reviewer for the valuable suggestions. In response to the query regarding the use of the BACE interface in smaller resistance arteries, we have explored its application in more accessible peripheral arteries, specifically the femoral artery. This investigation seeks to determine whether the BACE interface can offer meaningful insights into the physiological state of the aorta by monitoring electrical activity in smaller, more readily accessible arteries.

Figure R11. Monitoring of vasomotor activity in the femoral artery under normal and occlusion conditions using the BACE interface. **a**, Photograph of the BACE interface implanted on the femoral artery of a rabbit. **b**, Time-domain traces of electrophysiological signals recorded from the femoral artery before and after inducing proximal artery occlusion (abdominal aorta ligation). **c**, Power spectral density (PSD) curves of the signals recorded from the femoral artery before and after proximal artery occlusion. **d**, Quantitative analysis of the RMS power showing significant changes in the signal following occlusion, reflecting altered vasomotor function. Data in **c** are presented as mean values \pm SD. Box plots in **d** depict the data median (center line), upper and lower quartiles (box bounds), 1.5 times the interquartile range (whiskers) and outlier values beyond this range (circles). p values for power comparison before and after occlusion in **d**: $p = 1.1102 \times 10^{-4}$

¹⁶. (ns, : $p > 0.05$, * $p < 0.05$, ** $p < 0.01$, *** $p < 0.001$, **** $p < 0.0001$, Wilcoxon signed-rank test)

To this end, we implanted the BACE interface in direct contact with the femoral artery (**Figure R11a**). Initially, we recorded the electrophysiological signals of the femoral artery under normal conditions, followed by the induction of proximal artery occlusion through ligation of the abdominal aorta. The purpose of this experimental setup was to mimic a clinical scenario where downstream arteries are influenced by upstream vascular changes, a condition commonly encountered in cardiovascular pathologies. As shown in **Figure R11b**, significant attenuation was observed in the electrical activity of the femoral artery upon inducing proximal artery occlusion. We also performed a power spectral density (PSD) analysis of the signals within the 0-50 Hz frequency range (**Figure R11c**). The PSD curve before occlusion exhibited a clear distribution of signal energy across the low-frequency range. Following occlusion, the power spectrum revealed obvious alterations with a marked reduction in the low-frequency components. This observation was further supported by statistical analysis of RMS power of the femoral artery signals before and after the occlusion, which confirmed the suppression of vasomotor activity (**Figure R11d**). These findings underscore the BACE interface's capacity to detect and monitor vasomotor dysfunction not only in larger arteries like the aorta but also in smaller peripheral arteries, demonstrating the scalability and clinical relevance of our technology.

Figure R12. Demonstration of the extended wireless data transmission system based on our BACE interface. **a**, Schematic diagram of the wireless system architecture, comprising the power management module, signal acquisition, and wireless module. **b**, *In vitro* functional validation of

the wireless transmission system where a simulated 20 Hz, 100 μ V sine wave is transmitted via Bluetooth, showing the output on the PC screen. **c**, *In vivo* wireless recording of the vascular electrophysiological signals acquired from the aorta of a rabbit via the BACE interface.

Moreover, we are also working on the development of a wireless data transmission system to extend the capabilities of our bioelectronic interface (**Figure R12**). To validate the feasibility and reliability of this system, we first performed *in vitro* testing using a sine wave simulation input. This initial validation confirmed the reliable operation and performance of the wireless functionality. Then we have successfully demonstrated the wireless transmission of electrophysiological signals from the abdominal aorta in *in vivo* experiments, further reinforcing the potential of our system for real-time monitoring of vascular conditions. Looking forward, we aim to develop a fully-implantable wireless system for the continuous monitoring of vasomotor function in both laboratory and clinical settings, which takes the advantages of eliminating the need for external wiring and reducing the risk of infection. Additionally, a fully-implantable wireless system would facilitate continuous, real-time monitoring of vascular electrophysiology without interfering the subject's daily activities. The ability to remotely track changes in vasomotor activity would be invaluable for the early detection and management of vascular disorders, particularly for conditions like aortic aneurysms or arterial dissection where prompt intervention is critical. In conclusion, by integrating wireless transmission with a minimally invasive implantable solution, we believe our technology holds tremendous promise to be pushed further for widespread adoption in continuous, real-time monitoring of vascular health. This approach will not only provide valuable physiological information, but also enable personalized and dynamic interventions, driving significant advancements in vascular disease management.

Reviewer #4:

[1] Overall, very interesting work on an implantable, bioelectronic interface to monitor the bioelectrical activity of vasomotor functions. The manuscript is generally clear and well-written. I recommend a minor revision to address the below comments.

We sincerely thank the reviewer for the positive comments and we have carefully addressed all the points raised during the revision.

[2] The addition of PU to form the SF/PU composite decreased the young's modulus of SF. The authors state that a 10/6 ratio was used to achieve a young's modulus of <3 MPa to ensure mechanical compatibility with the aorta. Based on images of the device (Fig. 1d and Fig. S1) and the exploded-view in Fig. 1b, the overall device uses a solid thin film of PI. Thus the device likely has a young's modulus near 916 GPa similar to that measured for PI (Fig. S4).

a. Does the PI film dictate the modulus of the device?

b. If so, it would be helpful to measure the modulus of the entire device instead of only individual layers.

We sincerely thank the reviewer for the valuable comments. We fully agree that the mechanical properties of the polyimide (PI) film are a critical factor in determining the overall modulus of the device. In our design, PI was selected as the bottom substrate and top encapsulation layer for several important reasons. First, PI is an FDA-approved material widely recognized for its reliable and long-term biocompatibility in implantable medical devices. Second, its excellent compatibility with established MEMS and semiconductor fabrication processes enables high-resolution patterning and precise integration of electrode architectures. Third, PI offers robust mechanical strength and high fracture toughness, which are critical for maintaining structural integrity during device implantation and prolonged in vivo use. As the reviewer rightly points out, the relatively high Young's modulus of PI ($\sim 916.59 \pm 17.14$ MPa, as shown in **Supplementary Figure 4**) inherently limits its conformability on non-planar, three-dimensional biological surfaces. One possible strategy is to incorporate a softer supporting layer which will not compromise the overall structural integrity and bending stiffness of the device [Kim, D. H. et al. *Nat. Mater.*, **9**(6), 511-517 (2010)]. Inspired by this approach, we incorporated a bioadhesive silk fibroin (SF) layer as the outermost interface with biological tissue. Although the addition of the SF layer increases the overall thickness of the device, its low modulus ensures that it does not significantly contribute to the device's bending stiffness (**Supplementary Figure 5**), thereby preserving the desired flexibility and conformability. Furthermore, we have performed additional mechanical characterization on the effective modulus of the entire multilayer device, including the PI-based interface and the SF adhesive layer. This integrated structure exhibited a Young's modulus of 244.0 ± 54.17 MPa compared to the bare PI film (**Figure R13**). This reduction highlights the mechanical influence of the more compliant SF layer as well as structural heterogeneities within the multilayer assembly, and more accurately reflects the mechanical behavior of the device in its functional form. We have revised the relevant statements in the section "Design and Characterization of the BACE interface" of the manuscript.

Figure R13. Tensile performance of the BACE interface. Representative stress-strain curve for the BACE interface, with the cross indicating the breaking point. Reproducibility was ensured through measurements from $N = 3$ samples. Young's modulus is presented as mean \pm SD.

c. Is the silk substrate larger than the PI film and electrode layers? Is that what improves the adhesion to the 3 mm diameter tube? Since the interface consists of PI and PEDOT:PSS with direct contact to the tube/artery.

We sincerely thank the reviewer for the insightful comments and apologize for any confusion about the structural configuration of the device. The adhesive SF layer was deliberately designed and fabricated to be slightly larger than the underlying PI substrate and electrode regions. This geometric extension allows the peripheral edges of the SF to wrap beyond the PI boundaries and make secure adhesion with the external arterial surface. By adhering primarily at the outer margin, the SF layer ensures stable anchoring of the interface to the vessel without obstructing the electrode-tissue interface. We have accordingly revised the manuscript to more explicitly describe the role, size, and spatial positioning of the SF layer in relation to the PI and electrode components.

d. If the device does have a modulus larger than an artery/aorta, does wrapping the device around an artery constrict the expansion of the artery? If it's wrapped too loose, the electrical interface likely degrades.

We sincerely thank the reviewer for the important comments. With regard to the reliability and stability of the wrapping, while the overall modulus of the device is indeed higher than that of soft artery/aorta, the conformal attachment is achieved through localized adhesion rather than circumferential compression or tension. This allows the central region containing the electrodes to remain in compliant contact with the artery without exerting significant radial pressure. This stability was corroborated by our *in vivo* recordings. As shown in **Figure R14a**, the ultrasound images of the abdominal aorta during systole and diastole, along with the measured diameters, demonstrate that the BACE interface wrapping does not interfere with the vessel's natural expansion. Additionally, as depicted in **Figure R14b** and **R14c**, the electrical performance of the BACE interface, including both interfacial impedance and RMS amplitude of the basal signals, remained stable across multiple trials throughout the experiment. These results indicate that there was no detectable degradation in the electrical interface over time, even under physiological conditions involving continuous arterial pulsation and sustained tissue contact.

Figure R14. Reliability and stability of the wrapping around the artery. a, The ultrasound image of the abdominal aorta wrapped with the BACE interface indicates its normal expansion by the measured systolic and diastolic diameters. b, The interfacial impedance measured at 1 kHz during the three trials throughout the course of the experiment across $n = 3$ rabbits. c, The RMS amplitude of the baseline signals acquired from three trials on one rabbit as the experiment progresses. Data in b and c are presented as box plots which depict the data median (center line), upper and lower quartiles (box bounds), 1.5 times the interquartile range (whiskers) and outlier values beyond this range (circles). p values for impedance comparison: $p = 0.4508$ (Trial 1 vs. Trials 2), $p = 0.2154$ (Trial 1 vs. Trial 3), $p = 0.1258$ (Trial 2 vs. Trial 3). p values for amplitude comparison: $p = 0.1526$ (Trial 1 vs. Trials 2), $p = 0.4907$ (Trial 1 vs. Trial 3), $p = 0.0512$ (Trial 2 vs. Trial 3). (ns, $p > 0.05$, * $p \leq 0.05$, ** $p \leq 0.01$, *** $p \leq 0.001$, and **** $p \leq 0.0001$, Wilcoxon signed-rank test)

Moreover, to evaluate whether the device constrains natural arterial expansion, we conducted *in vitro* validation experiments in which the BACE interface was affixed to an elastic synthetic vessel (Figure R15a). The vessel was cyclically pressurized using a syringe to simulate systolic and diastolic changes in diameter (~ 0.3 mm), which closely resemble the physiological dynamics in rabbit abdominal aortas [Rahmani-Cherati, T. et al. *J. Tehran Heart Cent.*, 7(3), 128 (2012)]. Based on our *in vivo* ultrasound measurements in the abdominal aorta of rabbits, the typical diameter variation between systole and diastole is approximately 0.3 mm, corresponding to about 8-10% deformation relative to the resting diameter. This degree of deformation falls well within the stretchable range of our interface (Figure R13), confirming that the device is capable of accommodating physiological vessel pulsations without mechanical restriction. Taken together, these results from both *in vitro* and *in vivo* experiments provide evidence that the interface-tissue coupling remains intact and functionally robust throughout the experiment. The wrapping of the BACE interface does not interfere with the artery's natural vasomotor function and demonstrates sufficient stability to ensure that its electrical performance remains consistent over time.

Figure R15. *In vitro* experiment simulating vasoconstriction and vasodilation. **a**, Experimental setup showing the BACE interface wrapped around a synthetic vessel. Water is withdrawn and injected to simulate vessel contraction and dilation. Red arrows indicate the direction of fluid movement. **b**, Representative electrophysiological traces recorded during a quiescent period and during cyclic fluid pumping. The baseline signal remained stable with minimal amplitude drift when the interface is subjected to fluid-induced pressure. **c**, Impedance measurements during the phase of quiescence and fluid pumping. **d**, Quantitative comparison of baseline signal RMS amplitudes under quiescent and pumping conditions. Data in **c** and **d** are presented as mean values \pm SD. p values for data in **c** and **d**: $p = 0.9234$ (impedance comparison) and $p = 0.4656$ (baseline RMS comparison). (ns, : $p > 0.05$, * $p < 0.05$, ** $p < 0.01$, *** $p < 0.001$, **** $p < 0.0001$, Wilcoxon signed-rank test)

[3] Is there a reason 64 channels were used?

We sincerely thank the reviewer for the valuable comment. The reason to use 64 channels in our study was based on a careful consideration of the need to balance spatial coverage with signal resolution, as well as to minimize experimental complexity. Initially, we aimed to achieve a sufficient area of coverage with high resolution for the abdominal aorta of rabbits, which required a design of approximately 3 mm \times 15 mm to ensure the interface could wrap around the artery twice. Regarding electrode density, we experimented with various configurations and analyzed the signal cross-correlation coefficients between adjacent channels. In the final axial resolution configuration of 470 μ m, as shown in **Figure R16**, the signal similarity between the channel at the top-left corner

and its neighboring channels was higher, which demonstrates the retainment of some redundancy. This redundancy helps mitigate the risk of losing important signal information due to potential failures in individual channels, thereby increasing the overall reliability and robustness of the system. In contrast, the similarity between the corner channel and those with larger gaps was noticeably lower, indicating that distinct, valuable information was captured. Thus, after considering the requirements for both coverage area and signal resolution, the 64-channel configuration was selected to capture all localized variations in vasomotor activity. By enabling a more comprehensive sampling of physiological signals, this configuration facilitates thorough and robust analysis, ensuring that the intricate dynamics of vascular function are accurately monitored.

Figure R16. Analysis of signal cross-correlation coefficient between adjacent channels within the density configuration of 64 channels.

To demonstrate the advantages of using 64 channels, we also compared subsets of 4, 8, and 16 channels from the 64-channel array in a regular arrangement (**Figure R17**). The results indicate that with fewer channels, the distinction between different vasomotor activity states becomes less clear, reflecting a reduced ability to detect localized functional differences. In contrast, the 64-channel setup provides a richer representation of the vascular electrophysiological signals which is helpful in revealing the underlying structure of the data more robustly. This increased data richness can diminish the effects of noise and random fluctuations, resulting in more stable and interpretable low-dimensional embeddings (**Figure 2k**).

Figure R17. The t-SNE analyses on the feature datasets extracted from different numbers of channels. The subsets of 4, 8, and 16 channels are selected from the original 64 channels in a regular arrangement.

[4] For Fig. S14, was the same device moved during the *in vivo* experiment or were three devices implanted?

We sincerely thank the reviewer for the thoughtful comment. For the *in vivo* experiment shown in **Supplementary Figure 14**, the same device was indeed moved three times. This approach was intentionally chosen to maintain consistency throughout the experimental setup and to ensure the reliability of the results. By utilizing the same device for all three measurements, we were able to minimize the risk of introducing variability between devices, which could have potentially confounded our findings. Moreover, a critical aspect of using the same device multiple times was ensuring its ability to maintain stable performance after each repositioning. We have confirmed that the device could retain its adhesive properties, mechanical integrity, and electrical functionality across several cycles of movement and re-attachment. This validates the accuracy and reliability of our measurements, as it guarantees that the recorded signals are reflective of the electrophysiological state rather than being influenced by any degradation in the device's performance over time.

[5] Can it be confirmed that the changes in signal amplitude during the different studies (pharmacologic and stent) is not due to mechanical noise disturbing the electrodes?

We sincerely appreciate the reviewer's insightful comment regarding the potential influence of mechanical noise on our recorded signals. Firstly, we would like to emphasize that the electrical activity we measured in our study primarily originates from the autonomic nervous system, specifically via the postganglionic sympathetic nerve fibers located in the adventitia that innervate the vasculature. These fibers play a crucial role in modulating vasomotor function by regulating the contraction and relaxation of vascular smooth muscle cells through neurotransmitter release. When the vessels are exposed to pharmacologic agents or interventions such as stent implantation, these stimuli elicit the active response from the autonomic nervous system, resulting in dynamic adjustments in vasomotor tone. This neurogenic regulation involves a sophisticated interplay between sympathetic and parasympathetic pathways to maintain or restore vascular homeostasis across a range of physiological and pathological conditions. Therefore, the changes in signal amplitude recorded in our experiments are representative of this intrinsic neural modulation of vasomotor activity rather than artifacts caused by mechanical perturbations at the electrode-tissue interface.

Secondly, as demonstrated in our *in vitro* experiments (**Figure R15**), the BACE interface exhibits a stable baseline signal with negligible noise, drift, or fluctuations, even when subjected to simulated pulsatile mechanical disturbances mimicking vascular dynamics. Collectively, these findings reinforce our conclusion that the variations in signal amplitude observed during pharmacologic and stent studies genuinely represent autonomic neural activity governing vasomotor tone, and are not confounded by mechanical noise.

[6] Regarding the therapeutic effects of stimulation:

a. Was the stimulation only applied on day 14? More details on the stimulation timing/schedule are needed.

We thank the reviewer for the valuable comment. The stimulation by the BACE interface was indeed applied only on day 14 after stent implantation. This specific time point was selected based on clinical observations, as subacute in-stent thrombosis and arterial dysfunction typically arise

between days 2 and 14 following implantation. Additionally, early intimal hyperplasia begins immediately after the stent is placed and reaches its peak around day 14. Therefore, the day 14 stimulation corresponds to a critical phase in the vascular response post-implantation.

Furthermore, the Firehawk stent used in our study is a drug-eluting stent (DES) that gradually releases sirolimus, a potent immunosuppressive agent. The release of sirolimus is intended to inhibit cell proliferation and mitigate the risk of restenosis. However, as reported in previous research, the sirolimus-eluting stent (SES) can adversely affect endothelium-dependent vasomotor function in both large vessels and the microvasculature of the infarct-related coronary artery after reperfusion therapy. This dysfunction is associated with a reduction in myocardial vascular endothelial growth factor (VEGF) secretion, which is crucial for endothelial repair and vasodilation. Meanwhile, chronic exposure to sirolimus within the circulation for a duration of 2 weeks has the potential to cause a considerable accumulation of the drug in the vascular bed distal to the site of SES implantation, leading to selective aggravation of endothelial vasomotor dysfunction [Obata, J. E. et al. *J. Am. Coll. Cardiol.*, 50(14), 1305-1309 (2007)]. The 14-day time point chosen for stimulation in our study thus aligns with this critical period where the effects of drug accumulation and its impact on endothelial function are most pronounced. We have added a more detailed description of the stimulation timing and schedule in the main text and Methods section of the revised manuscript. We also recognize the importance of long-term stimulation at multiple time points, and this will be an integral component of our ongoing and future work.

b. The manuscript indicates it is ‘closed-loop’, so is the stimulation automated based on the recorded signals? If so, as shown in Fig. 4a, what was considered ‘dysfunction detection’?

We sincerely thank the reviewer for the insightful comment and apologize for any confusion regarding the stimulation approach. To clarify, while the stimulation in our study is not automated, the term ‘closed-loop’ in the manuscript is used to emphasize the synchronization between vasomotor function monitoring and electrical stimulation intervention. Specifically, in response to the vasomotor dysfunction observed in distal vessels following stent implantation, electrical stimulation was manually controlled based on predefined experimental parameters.

Regarding the ‘dysfunction detection’ mentioned in the manuscript, this concept is derived from the findings presented in **Figure 3**. Dysfunction detection refers to the observation of signal attenuation in the electrical signals detected at the distal end of the stented artery. These alterations in signal characteristics can serve as indicators of potential dysfunction in the vasomotor response, which guide our experimental approach for subsequent intervention.

c. In Fig 4g, how much time was there between the stimulation and the ‘after’ ultrasound image?

We thank the reviewer for the valuable comment. The Doppler ultrasound imaging was performed after the completion of all electrophysiological recordings and stimulation experiments. Specifically, the ultrasound was conducted following the ‘State 6’ as shown in **Figure 4e**. Consequently, there was an approximate 2-hour interval between the last stimulation (i.e., stim. #6) and the ultrasound imaging. This interval provides partial evidence that the electrical modulation induces a prolonged and sustained therapeutic effect, suggesting its potential for contributing to long-term improvements in vascular function.

d. Is the enhancement of electrical signal conduction and reduction of arterial stiffness a permanent

effect? Or does it dissipate with time? I'm not sure this requires additional animal studies, but comments should be added to the manuscript to clarify how long this effect was observed for after stimulation (minutes, hours, days, etc). (Side note: Future work on long-term stimulation and therapeutic effects would be interesting!)

We sincerely thank the reviewer for raising this important concern. Based on our current experimental data and findings, while we are not yet able to conclude that the enhancement of electrical signal conduction and the reduction of arterial stiffness have permanent effects, we have made efforts to investigate the sustained efficacy of electrical stimulation over longer time periods. As illustrated in **Figure R18**, following multiple intermittent stimulations, the recorded electrical signals remained stable for nearly 5 hours, indicating a prolonged modulation of vascular electrophysiological activity. In addition, arterial stiffness parameters β assessed approximately 5 hours after the final stimulation showed notable improvement compared to pre-stimulation baselines, suggesting an enhancement in vascular elasticity and compliance. These findings provide supportive evidence that electrical stimulation can yield therapeutic effects that extend well beyond the immediate post-stimulation window. We have added clarifying statements to the revised manuscript to explicitly specify the duration for which the effects were observed to persist following the final stimulation. Nevertheless, we fully recognize that the long-term persistence of these effects remains to be systematically characterized and constitutes a critical direction for future research.

Figure R18. Validation of the sustained effects of electrical stimulation. **a**, Electrophysiological response to intermittent 300 μA stimulation (indicated by red stars) over a prolonged period of time. **b**, The stiffness parameter β of the distal aorta before the first electrical stimulation and approximately 5 hours after the final stimulation. Data in **b** are presented as mean values \pm SD.

One of the key technical limitations currently restricting long-term stimulation and monitoring is the non-wireless nature of the present system. To address this, we are developing a second-generation BACE system with integrated wireless communication capabilities (**Figure R19a**). This upgraded system has already undergone preliminary validation both *in vitro* and *in vivo* settings, demonstrating its ability to reliably acquire and wirelessly transmit electrical signals with minimal distortion and high signal fidelity (**Figure R19b,c**). These results indicate that the new platform holds strong potential for enabling continuous, real-time monitoring and stimulation across extended durations, thereby overcoming the physical constraints associated with wired configurations. Our ongoing work is focused on further improving the system's miniaturization, biocompatibility, and energy efficiency to enable fully implantable, long-term applications in future studies.

Figure R19. Demonstration of the extended wireless data transmission system based on our BACE interface. **a**, Schematic diagram of the wireless system architecture, comprising the power management module, signal acquisition, and wireless module. **b**, *In vitro* functional validation of the wireless transmission system where a simulated 20 Hz, 100 μ V sine wave is transmitted via Bluetooth, showing the output on the PC screen. **c**, *In vivo* wireless recording of the vascular electrophysiological signals acquired from the aorta of a rabbit via the BACE interface.

Reviewer #5:

[1] The paper by Xiner Wang et al., entitled "Bioadhesive and Comformable Bioelectronic Interfaces for Vasomotoricity Monitoring and Regulation," addresses the feasibility of using an implantable sensor made with PEDOT:PSS/PI/Electrodes/PI/Silk Sub. for vasomotoricity monitoring and stimulation.

We sincerely thank the reviewer for the time involved in reviewing the manuscript and appreciate the valuable comments.

[2] Overall, the sensor design closely resembles existing structures, making its novelty somewhat limited. The study primarily presents preliminary findings on the sensor's potential for vasomotoricity monitoring and stimulation. However, in terms of technological maturity and practical usefulness, additional developments are necessary. These could include, for example, integrating wireless data transmission and power delivery, or identifying new findings and applications for the sensor.

We sincerely thank the reviewer for their thoughtful and constructive comments. We appreciate the opportunity to clarify and elaborate on the novelty, technical rationale, and ongoing advancements of our work.

First, we would like to emphasize the translational relevance and application-oriented innovation of our study. Our work addresses a critical and previously underexplored challenge in both vascular physiology research and clinical practice—the electrophysiological monitoring and modulation of vasomotor function mediated by the autonomic nervous system. While most existing vascular sensors have focused on hemodynamic parameters such as pressure or flow, few systems are designed to capture or modulate the bioelectrical signals underlying autonomic nerve control of vascular tone. Our proposed interface directly targets this gap by enabling simultaneous sensing and stimulation of autonomic nerve activity, thus offering a platform to deepen mechanistic insights into neurovascular regulation and investigate the potential therapeutics in vasomotor dysfunction.

Second, we would like to clarify that the resemblance to existing devices lies not in the interface geometry or application context, but rather in the use of standardized MEMS fabrication processes. The processes were selected for their exceptional compatibility with polyimide—a mechanically flexible, FDA-approved substrate known for its robustness and suitability for chronic implantation. Leveraging this fabrication pathway facilitates potential clinical translation while ensuring mechanical compliance with soft tissue surfaces. Moreover, to accommodate the complex three-dimensional morphology and dynamic motion of blood vessels, we integrated a bioadhesive silk-based substrate into the device design, which enables stable, conformal contact with the adventitia and reflects a material-level innovation tailored to vascular physiology.

Third, we fully agree with the reviewer that further development toward technological maturity is essential. To this end, we are actively developing a second-generation BACE system which integrates low-power Bluetooth and miniature implantable rechargeable batteries to enable wireless data transmission and power delivery. The preliminary prototype has successfully demonstrated real-time, wireless acquisition and transmission of electrical signals in both *in vitro* and *in vivo* settings without significant distortion (**Figure R20**). We are now optimizing the system's miniaturization, energy efficiency, and biocompatibility, aiming to enable fully implantable, closed-loop operation in future applications.

Figure R20. Demonstration of the extended wireless data transmission system based on our BACE interface. **a**, Schematic diagram of the wireless system architecture, comprising the power management module, signal acquisition, and wireless module. **b**, *In vitro* functional validation of the wireless transmission system where a simulated 20 Hz, 100 μ V sine wave is transmitted via Bluetooth, showing the output on the PC screen. **c**, *In vivo* wireless recording of the vascular electrophysiological signals acquired from the aorta of a rabbit via the BACE interface.

We appreciate the reviewer's valuable feedback and have revised our manuscript to more clearly highlight our new finding and applications based on the BACE interface.

[3] To improve the quality of the paper, it is crucial to emphasize the unique advantages of the sensor in both sensing and stimulation, especially when compared to existing invasive methods. Additionally, in terms of application, new findings or innovative applications should be included. We sincerely thank the reviewer for the insightful and constructive comments. We fully agree that it is essential to highlight the unique advantages of our interface in both sensing and stimulation, particularly in comparison to existing invasive methods currently used in clinical and experimental vascular assessments.

Conventional clinical techniques for vascular monitoring, such as ultrasound imaging and angiography, primarily capture macroscopic hemodynamic parameters (e.g., flow velocity, vessel diameter, or pressure) but lack the resolution or specificity to detect subtle electrophysiological changes involved in autonomic regulation of vasomotor function. Meanwhile, existing vascular

mechanosensors based on piezoresistive, capacitive, piezoelectric, or triboelectric modalities focus on detecting mechanical deformation, offering indirect measurements of vascular function without access to the underlying neural control mechanisms. In contrast, our interface directly targets the electrophysiological basis of vasomotor regulation and the neurovascular coupling that governs vasomotor tone. Moreover, our system not only enables real-time tracking of vasomotor responses but also allows for closed-loop neuromodulation, which is beyond the functional scope of current vascular diagnostic tools and mechanical sensors. To make this clearer, we have presented a comprehensive comparison of our interface with existing invasive methods in terms of principles, capabilities, and limitations in the point-by-point responses below. In addition, we also have revised our manuscript to better articulate the novel insights into the electrophysiological signatures associated with vasomotor dysfunction and also demonstrate its scalability and adaptability in new application scenarios.

Here are some comments that I think need to be properly addressed.

[4] 1) 33-35: Herein, we report a bioadhesive and conformable bioelectronic (BACE) interface with the incorporation of bioadhesive silk fibroin materials for high-fidelity monitoring and modulation of vasomotoricity.

> A supporting statement is needed that includes a clear rationale and supporting data to justify why the sensor enables high-fidelity monitoring. It is important to explain with specific reasons and provide relevant data or figures that demonstrate the extent to which high-performance monitoring is achievable with this sensor.

We sincerely thank the reviewer for the valuable comments. We fully recognize that a clear rationale and supporting data are essential to substantiate our claim of high-fidelity monitoring and modulation using the BACE interface. Below, we provide a detailed explanation supported by experimental evidence presented in the manuscript.

Firstly, our BACE interface is specifically engineered to achieve high signal fidelity through a combination of material, structural, and electrochemical design innovations. The integration of bioadhesive silk-based substrate enables intimate, stable, and conformal contact between the interface and the adventitia. The adhesive substrate, with a high interfacial toughness of approximately 100 N/m and a relatively low Young's modulus of less than 3 MPa, has a negligible effect on the overall bending stiffness of the device (from $1.6811 \times 10^{-9} \text{ N} \cdot \text{m}^2$ to $1.6854 \times 10^{-9} \text{ N} \cdot \text{m}^2$) while enhancing the interfacial adhesion. In *in vitro* aqueous environments, the interface maintained stable adhesion to the synthetic vessel for up to 60 days. This intimate coupling minimizes micro-motion artifacts and ensures consistent electrode-tissue contact *in vivo*, which is essential for reducing interface impedance and enhancing signal transduction. The electrodeposition of PEDOT:PSS further reduces the interface impedance by increasing the effective electrochemical surface area and enhancing charge transfer efficiency, thereby enabling more reliable detection of low-amplitude electrophysiological signals.

Secondly, regarding the supporting data to demonstrate the high-fidelity and high-performance monitoring capability, the electrochemical impedance spectroscopy (EIS) shown in **Figure 1e** exhibits low and stable interfacial impedance under deformation, with impedance values remaining below 3 k Ω at 1 kHz. This low impedance reflects a high charge transfer capacity at the electrode-tissue interface, which is critical for capturing weak endogenous bioelectrical signals with minimal distortion and contributes to the promising capability for high-fidelity electrophysiological

recordings [Wang, C. et al. *Sci. Adv.*, 8(20), eabo1396 (2022); Hazelgrove, B. et al. *Nat. Rev. Electr. Eng.*, 1-15 (2025)]. Additionally, **Figure 1f** shows the interface's enhanced charge storage capacity (CSC) (~100 times higher than that of bare gold), which indicates its ability to inject sufficient charge for effective stimulation without exceeding electrochemical safety limits. A high CSC is essential for bidirectional electrophysiological systems that require both recording and stimulation functions, as it ensures that stimulation pulses can depolarize target tissues without causing electrochemical damage [Jiang, Y. et al. *Science*, 375(6587), 1411-1417 (2022); Li, G. et al. *Adv. Mater.*, 34(15), 2200261 (2022)].

To validate the functional sensitivity of the BACE interface in capturing subtle vascular electrical activity *in vivo*, we designed a series of physiological perturbation experiments shown in **Figure 2**. Through graded administration of vasoactive drugs (e.g., norepinephrine and acetylcholine) and surgical sympathetic denervation, we simulated a spectrum of vasomotor states. Our recordings demonstrate that the BACE interface can distinguish fine-scale electrophysiological differences under varying degrees of vasoconstriction and vasodilation, suggesting a high degree of signal resolution and specificity. In **Figures 3b** and **3c**, we further characterize the *in vivo* electrical properties of the BACE interface. The interface exhibits a high yield of low impedance values at 1 kHz ($6.77 \pm 2.13 \text{ k}\Omega$), which is critical for achieving high-quality electrophysiological recordings. Moreover, the low background noise levels ($2.63 \pm 0.52 \mu\text{V}$) are significantly lower than the amplitudes of valuable electrophysiological signals. This characteristic ensures reliable discrimination of subtle electrophysiological signals from background noise, essential for high-fidelity recording. The combination of low impedance and minimal background noise validates the stability and fidelity of signal acquisition under physiological conditions, affirming the interface's functional competence post-implantation.

In summary, these results demonstrate that the BACE interface achieves high-fidelity monitoring through: (1) stable electrode-tissue coupling enabled by adhesion and mechanical compliance, (2) excellent electrochemical properties (low impedance, high CSC), and (3) reliable recording capability with low and stable background noise to detect and differentiate physiological vasomotor activity across a range of experimental conditions. We sincerely apologize for the overuse of emphatic adjectives such as "high-fidelity" in our manuscript and we have revised the abstract of the manuscript to provide clearer justification for the high-fidelity monitoring capabilities of the BACE interface. Thank you again for the constructive comment to help us strengthen the manuscript.

[5] 2) 36-38: The interface's conformal and stable adhesion to arteries in aqueous environments, combined with its exceptional electrical performance, enables precise recording of vasomotor states across varying conditions.

> Supporting details are needed regarding the exceptional electrical performance of the sensor. Please provide a clear rationale and be specific with the numerical data that demonstrate its outstanding electrical characteristics.

We sincerely thank the reviewer for the valuable comment. We would like to explain that this statement in the abstract is a summary of the experimental results presented in **Figures 1** and **2**, which demonstrate the mechanical and electrical performance of the BACE interface. Regarding the specific electrical characteristics, interfacial impedance and charge storage capacity (**Figure 1e, f**) are key parameters used to evaluate the electrical performance of bioelectronic interfaces [Zhou,

T. et al. *Nat. Mater.*, 22(7), 895-902 (2023)]. Our BACE interface maintains a low and stable impedance (below 3 k Ω at 1 kHz) even under mechanical strain. This is beneficial for minimizing signal attenuation and reducing noise interference, which ensures accurate electrophysiological recordings. Meanwhile, the interface exhibits a charge storage capacity approximately 100 times higher than that of the bare gold (Au) electrodes. This substantial increase in charge storage capacity allows the interface to effectively handle the required stimulation thresholds without exceeding electrochemical stability limits, thereby ensuring high-efficiency stimulation capabilities. Thus, the *in vitro* electrical characterization of the interface assesses its electrophysiological recording and stimulation capabilities, laying the foundation for its application in evaluating vasomotor function in specific settings. We sincerely apologize for the overuse of aggressive adjectives in our manuscript and we have revised the abstract and other relevant sections of the manuscript to more clearly explain the underlying principles and provide specific numerical data to support the exceptional electrical characteristics of the interface.

[6] 3) 38-40: In a stent implantation model, the BACE interface effectively monitors and identifies alterations in bioelectrical activity associated with abnormal vasomotor function, validated against clinical gold standards.

> Revise the sentence to make it clearer by providing specific examples (eg: clinical gold standards = arterial stiffness parameters in Ultrasound)

We sincerely thank the reviewer for this helpful suggestion. We have revised this sentence to “ In a stent implantation model, the BACE interface effectively monitors and identifies alterations in bioelectrical activity associated with abnormal vasomotor function, validated against the arterial stiffness parameters in ultrasound as clinical gold standards.”

[7] 4) 42-44: These findings offer new insights into vasomotor electrophysiology and present a promising approach for advancing vascular disease management in biomedicine.

> Provide specific examples and clarify the details.

We sincerely thank the reviewer for the valuable comment and we would like to provide further clarification to highlight the novel insights into vasomotor electrophysiology and the potential of our approach for advancing vascular disease management. This statement in the abstract summarizes the experimental results and future perspectives presented in **Figures 3** and **4** of the manuscript. As demonstrated in **Figure 3**, our interface is capable of detecting subtle, progressive changes in vasomotor function following stent implantation, which is characterized by a decrease in signal power and amplitude. These alterations in the electrical activity of the distal aorta reflect a decline in vasomotor function, which correlates with increased vascular stiffness and impaired elasticity as indicated by the stiffness parameter β . This ability to capture electrophysiological signatures directly associated with changes in vascular function is a significant advancement in the field, as current clinical imaging techniques like DSA and ultrasound provide structural data but do not offer direct insight into the electrophysiological mechanisms underlying vasomotor dysfunction.

Moreover, **Figure 4** illustrates the potential of the BACE interface for closed-loop regulation of vasomotor function. By simultaneously monitoring the distal aorta and applying electrical stimulation to the distal aorta, we demonstrated that electrical pulses (300 μ A) could effectively restore the amplitude of the bioelectrical signals from the distal aorta, suggesting a recovery of vasomotor function. Importantly, ultrasound measurements also revealed a reduction in arterial

stiffness after electrical stimulation, further confirming the improvement in vascular compliance. These results provide new insights into how electrical stimulation can modulate vasomotor function, presenting a promising approach for therapeutic interventions in vascular diseases where vasomotor dysfunction plays a significant role.

In conclusion, the BACE interface not only enables accurate monitoring of vasomotor dysfunction but also offers a novel platform for active modulation of vasomotor tone, making it a promising tool for advancing both research and clinical management of vascular diseases. We have revised the abstract and the relevant contents in Discussion section to make the conclusion more specific.

[8] 5) 71-74: While these methods provide detailed anatomical information crucial for diagnosing and treating vascular diseases, they offer limited insight into the dynamic regulation of vasomotor function under various physiological or pathological conditions.

> Provide more detailed explanations to clarify the reasons behind your statements. (why it provides limited insight?)

We sincerely thank the reviewer for the insightful comment. Among current clinical vascular examination methods, digital subtraction angiography (DSA), often considered one of the gold standards for assessing vascular lesions, offers dynamic information by showing blood flow within the vessels. However, its focus is primarily on vascular flow dynamics about the passage of contrast medium rather than on the electrophysiological changes associated with vasomotor function [Ruedinger, K. L. et al. *Am. J. Neuroradiol.*, 42(2), 214-220 (2021)]. While Doppler ultrasound (DUS) can provide dynamic, real-time insights into hemodynamic parameters, its efficacy is contingent upon operator experience and skill, potentially leading to inter-individual variability [Collins, R. et al. *Health Technol. Assess.*, 11(20), iii-184 (2007)]. Furthermore, it does not directly measure or monitor the electrophysiological regulation of vasomotor function. Computed tomography angiography (CTA) and magnetic resonance angiography (MRA) are excellent for providing high-resolution structural data, including information on vascular stenosis, patency, and large vessel morphology. However, these imaging techniques are primarily static, meaning they do not capture the dynamic functional changes of the vasculature. Additionally, these techniques are prone to artifacts and image distortion, which can further limit their accuracy and reliability in assessing functional parameters [Shwaiki, O. et al. *Int. J. Cardiovasc. Imaging*, 37, 3101-3114 (2021); Pollak, A. W. et al. *Circ. Cardiovasc. Imag.*, 5(6), 797-807 (2012)]. To sum up, these clinical modalities fundamentally differ from electrophysiological techniques in their principles and the nature of information they provide, leaving a gap in understanding how the neural regulation of autonomic nervous system influences overall vasomotor function.

Our BACE interface was designed to address the gap by enabling direct electrophysiological monitoring of vasomotor function. Through the collection of electrical signals from the adventitia, our system provides a real-time window into vasomotor tone and electrophysiological changes, which are not accessible through conventional imaging techniques. By detecting changes in electrical activity associated with vasoconstriction and vasodilatation, the BACE interface offers a novel approach to studying vasomotor function under various physiological and pathological conditions. We have added more detailed explanations in the revised manuscript and hope to better clarify the limitations of existing methods and the unique capabilities of our approach.

[9] 6) 79-91: Innovations in mechanical sensing technologies, including piezoresistive¹⁸, capacitive^{19,20}, piezoelectric^{21,22}, and triboelectric²³ sensors, have facilitated the measurement of vascular mechanical properties such as blood pressure, flow velocity, and pulse wave velocity. However, as a process governed by autonomic nerve activity, the vasomotor function relies fundamentally on electrophysiological events, which serve as the basis for neural signal transmission and the execution of physiological responses. Recent advances in implantable bioelectronic interfaces address these limitations by establishing direct contact with target tissues, enabling more precise measurement of physiological parameters compared to traditional methods. Despite these advancements, existing technologies are insufficient for directly and reliably recording high-quality vascular electrophysiological activity, which is essential for unraveling the electrophysiological mechanisms underlying the ANS regulation of vasomotor tone.

> Piezoresistive, capacitive, piezoelectric, and triboelectric sensors can also measure electrophysiological signals using different modalities. These sensors can directly and reliably record the signals, and even references 18, 20, and 21 allow wireless connectivity, providing convenient measurements.

> Please clarify your argument regarding the "recording of high-quality vascular electrophysiological activity." Why do you assert that your sensor provides high-quality recordings compared to the conventional sensors mentioned? If the sensing principles behind each technology are explained, along with a quantitative comparison of their pros and cons, and if the unique features of your developed sensor are highlighted with specific data or numbers, your argument will be much clearer.

We sincerely thank the reviewer for raising the important concerns. Blood vessel function critically depends on hemodynamics and the regulatory role of the autonomic nervous system at a macroscopic level, while electrophysiology constitutes the fundamental basis for monitoring neural regulatory activities. Therefore, our BACE interface is designed from an electrophysiological perspective to complement the insights provided by mechanical sensors. In response to the concerns raised, we have prepared a comparative overview of sensing principles, advantages and limitations of representative mechanical sensors (**Table R1**), and highlighted the unique features of our interface with supporting data.

We acknowledge that direct quantitative comparison between our BACE interface and mechanical sensors is challenging due to their different sensing principles. The sensors cited in **ref. 18-23** detect mechanical changes—strain, deformation, pressure, or motion—reflecting vascular function parameters such as blood pressure or flow velocity. However, their core sensing principles do not inherently capture electrophysiological signals associated with autonomic neural regulation of vasomotor tone. They lack the necessary signal specificity and sensitivity to reliably detect subtle electrical activity originating from autonomic nerves controlling vascular smooth muscle. Our proposed sensor directly interfaces with the adventitia to monitor electrophysiological signals generated by autonomic nerve activity. This direct electrical coupling benefits from the stable, low-impedance (<3 k Ω at 1 kHz) even under deformation, which minimizes signal loss and noise. The interface's high charge storage capacity (~100 times that of bare gold electrodes) supports safe and efficient electrical stimulation, enabling bidirectional neuromodulation. Therefore, our BACE interface achieves simultaneous electrical recording and stimulation, facilitating closed-loop regulation of vasomotor function, which is not achievable with conventional mechanical sensors. This combination of direct electrophysiological measurement, mechanical compliance, and

bidirectional functionality makes our BACE interface uniquely suited for unraveling the complex neurovascular mechanisms underlying vasomotor control.

Table R1. Comparison of the sensing principles and performance with representative vascular mechanical sensors.

Sensor Type	Sensing Principle	Advantages	Limitations	Ref.
Piezoresistive	Resistance change due to mechanical strain	 • Wireless monitoring of haemodynamics 	 • Susceptible to mechanical noise • Limited to physical measurements 	[18]
Capacitive	Fringe-field-capacitive sensing	 • Wireless monitoring of blood flow • Biodegradable 	 • Susceptible to environmental electromagnetic interference and external noise • Lack of direct electrical activity measurement 	[19]
		 • Wireless monitoring of diverse artery sizes and extents of occlusion 		[20]
Piezoelectric	Piezoelectric effect to measure electromotive force caused by mechanical force	 • Wireless monitoring of haemodynamics 	 • Susceptible to mechanical artefacts • Lack of capabilities for electrical recording and modulation 	[21]
		 • Biocompatibility 		[22]
Triboelectric	Contact electrification caused by mechanical motion	 • Self-powered • Bioresorbable 	 • Limited stability and repeatability • Only mechanical sensing 	[23]
BACE interface	Measure electrical signals generated by physiological processes	 • Sensitivity to subtle electrophysiological signals • Bidirectional, closed-loop recording and stimulation capability 	 • Lack of haemodynamics monitoring • Invasive 	This work

We have incorporated this comparative discussion into the revised manuscript to provide a clearer justification of our sensor’s capabilities. We greatly appreciate the reviewer’s valuable feedback to help us improve our work.

[10] 7) 100-106: This design allows for intimate and stable wrapping around blood vessels in aqueous environments, facilitating reliable, high-quality acquisition of vascular electrophysiological (VE) signals.

> Additional explanation is required to support this argument. What technologies enable this high-performance capability, and to what extent does the sensor achieve high performance? Specific numerical data is needed to demonstrate the level of performance.

We sincerely thank the reviewer for this valuable comment. The intimate and stable wrapping of the BACE interface around blood vessels in aqueous environments ensures conformal contact that minimizes micro-movements and significantly reduces tissue-electrode interfacial impedance. The incorporation of the conductive polymer PEDOT:PSS further enhances the interface’s electrical properties by increasing the effective electrochemical surface area and facilitating efficient charge transfer, resulting in improved signal quality and recording sensitivity. Thus, the interface achieves a low and stable impedance of less than 3 kΩ at 1 kHz *in vitro* under physiological deformation

(**Figure 1e**) and maintains impedance values below 10 k Ω at 1 kHz *in vivo* (**Figure 3b**), which is critical for reducing signal attenuation and noise interference. To further validate the quality and reliability of *in vivo* signal acquisition, we assessed the low background noise levels across multiple experimental subjects. The results, presented in **Figure 3c**, demonstrate consistently minimal noise levels ($2.63 \pm 0.52 \mu\text{V}$) compared to the vascular electrical signal amplitude, highlighting the robustness and reproducibility of the BACE interface in capturing vascular electrophysiological signals under physiological conditions. This quantitative assessment reinforces the interface's capability for stable, high-quality signal recording essential for monitoring subtle vasomotor dynamics. We have incorporated these data and clarifications into the revised manuscript to address the reviewer's concerns.

With its high-fidelity and high-resolution recording capabilities, the BACE interface enables precise monitoring of vasomotor function in distal arterial segments following the implantation of the metal stent with performance validated against the stiffness parameter, which is the clinical gold standard in arterial assessment.

> Evidence is needed to support the claim of high fidelity. In terms of high resolution, data is needed to show how the 64 implemented channels affect the results when compared to using fewer or more than 64 channels.

We sincerely thank the reviewer for the valuable comments. The high-fidelity electrophysiological recording capability of our BACE interface is fundamentally supported by its electrical properties, including low and stable interfacial impedance, as demonstrated in **Figures 1e**. The characteristic is essential for minimizing signal attenuation and noise interference, thereby enabling accurate capture of subtle bioelectrical signals. To further validate this high fidelity, we performed physiological perturbation experiments (**Figure 2**) involving graded administration of vasoactive drugs and surgical sympathetic denervation to simulate a range of arterial vasomotor states. Our interface successfully distinguished these varying degrees of vasomotor tone through electrophysiological signal variations, underscoring its ability to reliably detect nuanced physiological changes.

The reason to use 64 channels in our study was based on a careful consideration of the need to balance spatial coverage with signal resolution, as well as to minimize experimental complexity. Initially, we aimed to achieve a sufficient area of coverage with high resolution for the abdominal aorta of rabbits, which required a design of approximately 3 mm \times 15 mm to ensure the interface could wrap around the artery twice. Regarding electrode density, we experimented with various configurations and analyzed the signal cross-correlation coefficients between adjacent channels. In the final axial resolution configuration of 470 μm , as shown in **Figure R21**, the signal similarity between the channel at the top-left corner and its neighboring channels was higher, which demonstrates the retainment of some redundancy. This redundancy helps mitigate the risk of losing important signal information due to potential failures in individual channels, thereby increasing the overall reliability and robustness of the system. In contrast, the similarity between the corner channel and those with larger gaps was noticeably lower, indicating that distinct, valuable information was captured. Thus, after considering the requirements for both coverage area and signal resolution, the 64-channel configuration was selected to capture all localized variations in vasomotor activity. By enabling a more comprehensive sampling of physiological signals, this configuration facilitates thorough and robust analysis, ensuring that the intricate dynamics of vascular function are accurately monitored.

Figure R21. Analysis of signal cross-correlation coefficient between adjacent channels within the density configuration of 64 channels.

To demonstrate the advantages of using 64 channels, we also compared subsets of 4, 8, and 16 channels from the 64-channel array in a regular arrangement (**Figure R22**). The results indicate that with fewer channels, the distinction between different vasomotor activity states becomes less clear, reflecting a reduced ability to detect localized functional differences. In contrast, the 64-channel setup provides a richer representation of the vascular electrophysiological signals which is helpful in revealing the underlying structure of the data more robustly. This increased data richness can diminish the effects of noise and random fluctuations, resulting in more stable and interpretable low-dimensional embeddings (**Figure 2k**). We have incorporated the supporting data into the revised manuscript to better clarify and substantiate these claims.

Figure R22. The t-SNE analyses on the feature datasets extracted from different numbers of channels. The subsets of 4, 8, and 16 channels are selected from the original 64 channels in a regular arrangement.

[11] 8) 109-111: Our approach holds significant potential for elucidating the electrophysiological mechanisms underlying vasomotor function and advancing both the clinical diagnosis and therapeutic intervention of vascular diseases.

9) 124-126: This innovation not only deepens the understanding of the bioelectrical mechanisms underlying vasomotor dynamics but also introduces promising therapeutic strategies for addressing vasomotor dysfunction.

> If specific applications and supporting evidence are added to the above statements, the overall quality of the paper would be significantly improved.

We sincerely thank the reviewer for this valuable suggestion and we are sorry for lack of clarity. We would like to explain that these two sentences summarize the monitoring and regulation of vasomotor dysfunction presented in **Figures 2-4**. Initially, we validated the recording capability of our BACE interface and tested our hypothesis regarding autonomic nervous system-mediated regulation of vasomotor function by simulating varying degrees of arterial vasomotor states. This was achieved through graded administration of vasoactive drugs at different doses combined with surgical sympathetic denervation, allowing us to demonstrate that the interface can reliably capture subtle electrophysiological changes corresponding to different vasoconstriction and vasodilation levels (**Figure 2**). Building upon this validation, we selected the stent implantation model as a clinically relevant application scenario because vascular stenting is widely used in the treatment of occlusive arterial diseases but often leads to vasomotor dysfunction and restenosis, of which the underlying electrophysiological mechanisms cannot be resolved by conventional methods. The monitoring of our BACE interface demonstrated distinguishable electrophysiological changes associated with vasomotor impairment distal to the stent, capturing signal attenuation and reduced vasomotor tone over time (**Figure 3**). Furthermore, the interface's bidirectional functionality enabled closed-loop electrical stimulation at the stent site, which successfully restored vasomotor electrophysiological signals and improved vascular compliance of the distal aorta, as confirmed by ultrasound-based arterial stiffness measurements (**Figure 4**). These experimental results concretely support the potential of our approach not only to deepen understanding of the bioelectrical mechanisms underlying vasomotor dynamics but also to offer promising therapeutic strategies for vascular disease intervention. We hope that incorporating this detailed explanation enhances the clarity of the manuscript, and we sincerely appreciate the reviewer's constructive feedback.

[12] 10) 114-126: Overview of the BACE interface Section

> If the electrophysiological signal sensing principles of the developed sensor, along with the mechanism of dysfunction remission through electrical stimulation, are explained, it would be easier for beginners to understand.

We sincerely thank the reviewer for this valuable suggestion. Our sensor is designed to directly measure vascular electrophysiological signals arising from the autonomic nervous system's regulation of vascular smooth muscle tone. These signals reflect the electrical activity generated by sympathetic and parasympathetic nerve fibers innervating the vessel wall. By establishing intimate, low-impedance electrical contact with the vascular surface through the bioadhesive silk-based substrate, our interface is able to capture these subtle bioelectrical signals, thereby enabling precise monitoring of vasomotor function in real time.

Regarding the mechanism by which electrical stimulation may alleviate vasomotor dysfunction, our approach is inspired by well-established clinical neuromodulation modalities such as spinal cord stimulation (SCS) and deep brain stimulation (DBS), which provide valuable conceptual frameworks. SCS is understood to modulate pain by activating large diameter afferent fibers that inhibit nociceptive signal transmission within the spinal dorsal horn [**Sdrulla, A. D. et al. *Pain Pract.*, 18(8), 1048-1067 (2018); Heijmans, L. et al. *Postgrad. Med.*, 132(sup3), 17-21 (2020)**]. The consensus regarding the mechanism of DBS is that high-frequency stimulation modulates neural activity across afferent and efferent brain regions to restore function and exert therapeutic effects [**Lozano, A. M. et al. *Nat. Rev. Neurol.*, 15(3), 148-160 (2019); Cagnan, H. et al. *Nat. Biotechnol.*, 37(9), 1024-1033 (2019)**]. Building on these insights, we hypothesize that electrical

stimulation applied to autonomic nerve fibers innervating the artery may help restore appropriate neural conduction and rebalance sympathetic and parasympathetic inputs. Through modulating this neural activity with targeted electrical pulses, we anticipate potential improvements in vasomotor tone and vascular elasticity. While this remains a working hypothesis informed by analogous neuromodulation paradigms, our experimental findings provide encouraging initial evidence supporting this mechanism. To enhance clarity and accessibility for readers less familiar with the field, we have revised the manuscript to include a brief explanation of the electrophysiological signal sensing principles of our developed sensor, as well as the mechanisms by which electrical stimulation can alleviate vasomotor dysfunction.

[13] 11) 192-194: Notably, the ratio of rising edge duration to falling edge duration within a single signal cycle exhibited characteristic patterns dependent on the vasomotion state.

> Does all the observed data reflect the same phenomenon? It would be helpful to explain how many data points were collected and include an analysis of how the trend changes with varying doses. Additionally, if possible, it would be beneficial to discuss the physiological significance of these findings.

We sincerely thank the reviewer for the insightful comments. We would like to clarify that all the observed data from three rabbits reflect the same phenomenon. To provide a more comprehensive and robust analysis, we have revised **Figure 2** by supplementing the data with time-domain waveforms and time-frequency spectrograms corresponding to different doses of vasoactive agents from one rabbit. The corresponding results from the other two rabbits under the same experimental conditions are shown in **Figure R23**. We have also updated main **Figures 2d** and **2h** to reflect the aggregated experimental results from all three rabbits. With increasing doses of these vasoactive agents, both vascular contraction and dilation are enhanced, as evidenced by the progressively larger amplitudes and higher low-frequency power of the vascular electrical signals recorded via the BACE interface. On the other hand, marked differences can be observed from the enlarged view of the signals under different pharmacological influences.

Regarding the physiological significance, the observed variations in the ratio of rising-to-falling edge durations correspond to distinct vasomotor states induced by graded administration of norepinephrine (NE) and acetylcholine (ACh). These changes reflect alterations in the contraction and relaxation dynamics of vascular smooth muscle, which are tightly regulated by the autonomic nervous system. Specifically, increasing doses of NE, a sympathetic agonist, tend to prolong the falling phase indicative of vasoconstriction, whereas ACh, a parasympathetic neurotransmitter, promotes vasodilation manifesting as relatively prolonged rising phases. This ratio thus serves as a sensitive electrophysiological biomarker that captures the balance between sympathetic and parasympathetic regulation of vasomotor tone. We hope these revisions and explanations enhance the completeness and interpretability of our analysis and thank again for the important questions.

Revised Figure 2. Evaluation of precise vasomotor electrophysiology recording. **a**, Schematic illustrating the exogenous administration of norepinephrine (NE) to the artery, inducing vasoconstriction. **b**, Representative channel depicting electrical responses before and after the administration of different doses of NE (0.1, 0.2, and 0.4 mL) in an anaesthetized rabbit. **c**, Enlarged view of the VE signals following NE administration. **d**, Relative amplitude of signals before and after the application of varying doses of NE from $n = 3$ rabbits. **e**, Schematic showing the exogenous delivery of acetylcholine (ACh) to the artery that leads to vasodilatation. **f**, Representative channel showing electrical responses before and after the delivery of different doses of ACh (0.1, 0.2, and 0.4 mL) in an anaesthetized rabbit. **g**, Enlarged view of the VE signals after ACh delivery. **h**, Relative amplitude of signals before and after applying different doses of ACh from $n = 3$ rabbits. **i**, Schematic depicting sympathetic ganglion transection, resulting in vasomotor suppression. **j**, Representative channel showing electrical responses before and after desympathize in an anaesthetized rabbit. **k**, Visualization of VE signal feature space after dimensionality reduction via t-SNE, signifying varying vasomotor states induced by different interventions. The signals in **b**, **f**, and **j** are displayed as time-domain traces (top) and frequency-domain spectra (bottom). The dashed boxes in **b** and **f** denote the time windows from which **c** and **g** are derived. The time intervals between consecutive peaks and valleys are represented by t_r and t_f . Data in **d** and **h** are presented as mean values \pm SD from $n = 3$ rabbits.

Figure R23. Vascular electrical responses evoked by varying doses of different vasoactive agents. The representative time-domain waveforms and time-frequency spectrograms of the VE signals recorded under basal condition and after NE and ACh delivery from two rabbits (left) highlight the dose-dependent effects on vasomotor function. The zoomed-in view of the time-domain waveforms (right) provides a more detailed comparison of the amplitude and rhythm of the responses induced by different drugs at different doses.

[14] 12) 204-207: Denervation leads to a significant reduction in VE activity, while compensatory mechanisms involving intact ganglia and circuits help maintain partial arterial function, allowing for the detection of weak electrical signals (Fig. 2j and Fig. S10).

> In this case, it would be helpful to include whether electrical stimulation leads to an increase in the damaged vascular electrophysiological (VE) activity, as this would enhance the novelty of the paper.

We sincerely thank the reviewer for raising this insightful question. We have conducted additional experiments applying electrical stimulation to the abdominal aorta following sympathetic ganglion denervation. As shown in **Figure R24**, stimulation at currents of 100, 300, and 500 μA elicited a modest recovery of vascular electrophysiological (VE) signals compared to the post-denervation baseline; however, signal amplitudes did not fully return to levels observed prior to nerve transection. This partial restoration may be attributed to compensatory neural mechanisms involving residual intact ganglia and alternative autonomic pathways that maintain some degree of vasomotor function despite the disruption of primary sympathetic input. The incomplete recovery likely reflects the complex and distributed nature of autonomic innervation, wherein electrical stimulation can transiently enhance neural activity but cannot entirely replicate the lost physiological signaling. Additionally, factors such as altered receptor sensitivity or vascular remodeling after denervation may further constrain the extent of functional recovery. These findings elucidate both the therapeutic potential and inherent limitations of electrical stimulation for modulating impaired vasomotor function, offering valuable insights into the neurovascular dynamics following autonomic injury. We have incorporated these results and interpretations into the revised manuscript to enrich the discussion and the novelty of our approach.

Figure R24. Effects of electrical stimulation with different current intensities to the aorta after denervation. **a**, Representative time-domain waveforms and time-frequency spectrogram of the damaged vascular electrophysiological activity before and after electrical stimulation with intensities of 100, 300, and 500 μA . **b**, Quantitative analysis of vascular electrophysiological signal power under different conditions. Box plots in **b** depict the data median (center line), upper and lower quartiles (box bounds), 1.5 times the interquartile range (whiskers) and outlier values beyond this range (circles). p values for power comparison: $p = 2.848 \times 10^{-4}$ (Normal vs. Denervation), $p = 0.0053$ (Normal vs. 100 μA stimulation), $p = 0.0449$ (Normal vs. 300 μA stimulation), $p = 0.0028$ (Normal vs. 500 μA stimulation). (* $p < 0.05$, ** $p < 0.01$, *** $p < 0.001$, **** $p < 0.0001$, Friedman test with Dunn's post hoc test).

[15] 13) Fig2.f : The basal condition appears small when compared to Fig. 2b and 2j. Is there a reason for this?

We sincerely thank the reviewer for raising this insightful concern. The observed differences in baseline signal amplitudes between **Figures 2b** and **2j** can be attributed to several factors, including inherent inter-individual variability among the rabbits and variations in the depth of anesthesia during experiments, which are known to influence autonomic nervous system responsiveness

[Bankenahally, R. et al. *Bja Education*, 16(11), 381-387 (2016)]. Such baseline variability is a common and expected phenomenon in *in vivo* electrophysiological recordings. Importantly, our analysis focuses on the relative changes in signal power elicited by pharmacological or surgical interventions, which serve as more reliable and physiologically relevant indicators of vasomotor states. We have emphasized this point in the revised manuscript to clarify the interpretation of our results.

[16] 14) 227-232: The BACE interface conformed seamlessly to the aorta, ensuring intimate contact and reducing tissue-electrode impedance, with all 64 recording sites demonstrating impedance values below 10 k Ω (Fig. 3b).

> Is a contact impedance of 10k Ω considered good? Why or why not? What is the impedance level for other technologies? Including references or data would help clarify the explanation.

We sincerely thank the reviewer for raising this important question. In the context of bioelectronic interfaces, the tissue-electrode interface impedance is a critical parameter that directly influences the quality of recorded signals [Hazelgrove, B. et al. *Nat. Rev. Electr. Eng.*, 1-15 (2025)]. As the increased electrochemical impedance is expected to enhance the thermal noise, thus reducing the signal-to-noise ratio (SNR). Interface impedance is influenced by multiple factors, including the electrode material's intrinsic conductivity, surface morphology and roughness, electrochemical properties (e.g., double-layer capacitance and charge transfer resistance), as well as the quality of contact between the electrode and the biological tissue, which is further modulated by mechanical conformity and bioadhesion at the interface [Lewis, C. M. et al. *Adv. Healthc. Mater.*, 13(24), 2303401 (2024)].

The impedance of the BACE interface, with values below 10 k Ω at 1 kHz across all 64 recording sites, reflects a favorable electrical coupling with the vascular tissue, especially considering the challenging *in vivo* environment characterized by motion and fluid dynamics. When compared to other state-of-the-art flexible bioelectronic platforms in the field of neural and cardiovascular research, the performance of our BACE interface remains competitive. For example, the flexible electrodes coated with PEDOT:PSS for tracking neuronal ensembles have been reported to achieve interface impedances of 54 ± 16 k Ω at 1 kHz [Yasar, T. B. et al. *Nat. Commun.*, 15(1), 4822 (2024)]. Another ultraconformable cuff implant designed for interfacing of peripheral nerves shows an *in vivo* impedance of 41.64 ± 128.06 k Ω at 1 kHz [Carnicer-Lombarte, A. et al. *Nat. Commun.*, 15(1), 7523 (2024)]. Thus, our BACE interface with an *in vivo* impedance of 6.77 ± 2.13 k Ω at 1 kHz can meet the requirement for high-quality electrophysiological signal acquisition. Whereas piezoresistive, capacitive, piezoelectric, and triboelectric sensors, which primarily measure mechanical rather than electrical signals, do not have comparable direct electrical interface impedance metrics, making direct comparison challenging.

This exceptional electrical performance enabled reliable acquisition of vascular electrical activity associated with vasomotor states, yielding recordings characterized by low baseline noise and stable signal amplitudes (Fig. 3c).

> In terms of baseline noise and signal amplitudes, how do other technologies compare?

We sincerely thank the reviewer for raising this important concern. The importance of low baseline noise cannot be overstated in the accurate detection of subtle vascular electrical activities associated with vasomotor states. Our study showcases the performance of our BACE interface in this regard,

characterized by consistently low baseline noise levels ($2.63 \pm 0.52 \mu\text{V}$). This attribute is paramount for ensuring accurate detection and analysis of vascular electrophysiological signals. Specifically, our background noise levels are not only significantly lower than those reported for recent advancements in ultra-conformal skin electrodes ($\sim 10 \mu\text{V}$) [Zhao, Y. et al. *Nat. Commun.*, **12**(1), 4880 (2021); Kim, H. et al. *Sci. Adv.*, **10**(3), eadk5260 (2024)], but also approaching the noise standards documented for state-of-the-art peripheral nerve microelectrodes (noise std: $\sim 1.9\text{-}3.7 \mu\text{V}$) [Dong, C. et al. *Nat. Mater.*, **23**(7), 969-976 (2024)]. This underscores our low baseline noise levels meet the stringent requirements for high-quality electrophysiological recordings, ensuring precision and reliability in physiological monitoring. Signal amplitudes depend on the type of signals measured, making direct comparisons between different signals less meaningful. In addition, comparing baseline noise and signal amplitudes with other mechanical sensors and clinical vascular examination methods is challenging due to the inherent differences in measured quantities. We hope this quantitative evidence strongly supports the efficacy and suitability of our sensor for its intended application.

[17] 15) 238-240: In contrast, our multi-channel BACE interface enabled high-resolution tracking of electrophysiological changes in the distal aorta, directly reflecting vasomotor status.

> What are the effects of using multiple channels? An explanation of the correlation between the number of channels and their characteristics is needed, along with a clarification of what high-resolution tracking means. Additionally, data and further explanation should be provided to support this.

We sincerely thank the reviewer for the valuable comments. The utilization of multiple channels in our study serves several critical purposes. Firstly, employing a 64-channel configuration allows us to achieve comprehensive spatial coverage along with high signal resolution. This setup was carefully chosen to accommodate the need for capturing localized variations in vasomotor activity across the abdominal aorta of rabbits. The configuration of approximately $3 \text{ mm} \times 15 \text{ mm}$ was specifically designed to wrap around the artery twice, ensuring intimate and stable electrode-tissue contact for accurate signal acquisition.

Regarding the correlation between the number of channels and their characteristics, our analysis in the final axial resolution configuration of $470 \mu\text{m}$ (**Figure R21**) demonstrates that adjacent channels within the array exhibit higher signal similarity, indicating redundancy that enhances the system's reliability. This redundancy helps mitigate potential signal loss due to individual channel failures, thus maintaining robustness in signal acquisition. Conversely, channels with greater spacing show lower signal similarity, highlighting their ability to capture distinct and valuable physiological information.

High-resolution tracking, in this context, refers to the capability of the 64-channel array to discern subtle variations in vasomotor activity with precision. Our comparative analyses using subsets of 4, 8, and 16 channels (**Figure R22**) underscore that fewer channels result in diminished differentiation between different vasomotor states, indicating reduced sensitivity to localized functional differences. In contrast, the 64-channel setup provides a richer representation of vascular electrophysiological signals, facilitating more detailed and accurate tracking of dynamic vascular responses. The enhanced data richness achieved with the 64-channel configuration minimizes the impact of noise and random fluctuations, contributing to stable and interpretable low-dimensional embeddings (**Figure 2k**).

In summary, the use of multiple channels not only enables comprehensive spatial coverage and high signal resolution but also enhances the system's ability to track dynamic vascular responses with precision and robustness. We have incorporated this supporting data and expanded explanation into the revised manuscript to better clarify the relationship between channel number and data quality. We sincerely appreciate the reviewer's constructive feedback, which has greatly strengthened our work.

[18] 16) 295-298: These results suggest that electrical stimulation may enhance electrical signal conduction in dysfunctional vessels, thereby improving vascular elasticity and compliance. This presents a promising therapeutic strategy for mitigating vasomotor dysfunction.

> This appears to be a good outcome

We sincerely thank the reviewer for the positive comment. We are encouraged that the potential of electrical stimulation to enhance signal conduction and improve vascular elasticity and compliance is viewed as promising. This finding not only supports the therapeutic relevance of our approach but also provides a foundation for future investigations into closed-loop neuromodulation strategies aimed at restoring vasomotor function. We appreciate the reviewer's recognition of this outcome, which reinforces the significance of our work.

Point by point response (comments in black and responses in blue):

Reviewer #2:

[1] The authors have mostly addressed my comments.

We sincerely thank the reviewer for their thoughtful comments and valuable feedback. We remain committed to improving the manuscript and are grateful for the further suggestions to enhance its clarity.

[2] With my improved understanding from the authors' response, I have two remaining comments relating to Fig. 4. 1) Interpretation of the "states" elicited from the subsequent input stimuli. Why does a neuromuscular stimulation result in a sustained relaxation(?) after the input stimulus? I do recognize the apparent improvement in flow, as evidenced in Fig 4g, but I would anticipate that the signal would go back to baseline after some time for a neuromuscular stimulation. Instead, it appears that each stimulus application results in concurrent increases in the electrophysiological response that is sustained for some time past the authors recording. If this is a stimulation of the local junctions with vascular smooth muscle, why does the muscle not constrict(?) again after the stimulation period ends? Please provide some interpretation of these data and the sustained changes in the text.

We sincerely thank the reviewer for the insightful comments. We also apologize for any ambiguity regarding the interpretation of the electrophysiological signals and vascular response following stimulation. First, as illustrated in **Figures 2 and 3**, our recorded electrophysiological signals capture changes in vasomotor activity under various conditions, including enhancement induced by vasoactive agents and attenuation following stent implantation. We would like to clarify that these signals reflect the overall vasomotor capacity—that is, its intrinsic ability to contract and relax—rather than indicating a singular episode of vasodilation or vasoconstriction. The observed sustained increase in signal amplitude following neuromuscular (or electrical) stimulation represents a recovery or enhancement of this vasomotor competence rather than a persistent state of dilation or contraction.

Second, **Figure 4g** is intended to show changes in arterial stiffness parameters as measured by vascular ultrasound, rather than immediate alterations in blood flow. While blood flow dynamics typically evolve in the later stages post-stenting, arterial stiffness serves as an important early biomarker for in-stent restenosis risk [**Prskalo, Z. et al. BMC Cardiovasc. Disord., 16, 1-6 (2016)**]. The inclusion of arterial stiffness measurements in **Figure 4g** serves to provide an independent validation of the changes observed in the electrophysiological recordings. The concurrent improvement in both arterial elasticity and electrophysiological amplitude strengthens the interpretation that electrical stimulation enhances overall vasomotor function instead of inducing a simple transient mechanical response.

Lastly, our stimulation strategy was inspired by previous studies demonstrating that localized electrical stimulation of arterioles can effectively elicit vasomotor responses through mechanisms involving microvascular ion channels and signaling pathways [**Figuroa, X. F. et al. Am. J. Physiol. Heart Circ. Physiol., 293(3), H1371-H1383 (2007)**; **Sgouralis, I., & Layton, A. T. Am. J. Physiol. Renal Physiol., 303(2), F229-F239 (2012)**; **Hald, B. O. et al. Pflügers Arch., 467, 2055-2067 (2015)**]. In particular, ion channels involved in vasomotor tone regulation have been shown to facilitate long-range signal propagation across the vascular wall, often manifesting as site-specific

vasoconstriction accompanied by conducted vasodilation that spreads nondecrementally through intercellular gap junctions. In our study, the sustained elevation in electrophysiological response after stimulation may be attributable to the persistent effect of electrical stimulation on the fast electrical signaling through intercellular gap junctions within the vascular wall, leading to downstream modulation of vasomotor response. This mechanistic interpretation also supports the potential role of early and repeated electrical stimulation in preventing the progressive suppression of vasomotor responsiveness in the distal aorta after stent deployment. We appreciate the reviewer's constructive comments and have added a discussion on the interpretation of the sustained changes in the revised manuscript.

[3] 2) Closed-loop system. In Fig. 4a, if the applied stimulus is not automated, as indicated in the authors' response, then calling this system "closed-loop" is somewhat of an exaggeration. I do not think this is consequential for the novelty of the manuscript, as the data collection suggests that a closed-loop system could be constructed, but the authors should adjust their wording to clarify.

We sincerely thank the reviewer for the valuable comment and apologize for the exaggerated use of the term "closed-loop" in the manuscript. As correctly noted, the current system does not incorporate automated, real-time feedback to adjust the stimulation parameters based on incoming physiological signals. Therefore, to more accurately reflect the current design, we have revised the terminology to "bidirectional system" throughout the manuscript. This revised description emphasizes the system's dual capacity to record electrophysiological signals and deliver targeted electrical stimulation. We appreciate the reviewer's helpful suggestion in guiding this clarification and the corresponding revisions have been made in both the text and figure captions to ensure consistency and accuracy.

Reviewer #3:

The authors provided an authoritative and clear response to my concerns.

We sincerely thank the reviewer for the thoughtful comments to greatly contribute to the improvement of the manuscript and we are truly grateful for the encouraging feedback.

Reviewer #4:

Looks good, recommend publishing

We sincerely thank the reviewer for the positive comments. We are truly grateful for the constructive feedback throughout the review process, which have greatly helped us improve the quality and clarity of the manuscript.

Reviewer #5:

Since spatial resolution appears to be a critical system specification, it would strengthen the manuscript if the authors could also discuss the potential effects of varying spatial resolutions. Currently the revised version only compares data at 4,8,16 channels under the same spatial resolutions.

We sincerely thank the reviewer for this valuable comment. We also apologize for any confusion caused by our previous explanation regarding the spatial resolution settings. In our previous revision, we selected electrode channels within a fixed electrode array contour, varying the axial spatial resolution by adjusting the inter-electrode distance along the longitudinal direction. In other words,

for each spatial resolution condition, we extracted a subset of channels corresponding to the desired spacing within the same total electrode length, thereby enabling a fair comparison of signal quality across different resolutions. We agree with the reviewer that spatial resolution is a critical parameter, and we have accordingly clarified the axial spatial resolution (defined by the electrode spacing) in the revised manuscript (**Figure R1**). To further strengthen the discussion, we have also added the 32-channel recording results in the supplementary materials. Notably, as the axial spatial spacing decreases, the system’s ability to distinguish different levels of vasomotor activity improves. The 64-channel configuration was ultimately chosen not only to achieve high spatial resolution, but also to introduce a degree of redundancy, thereby minimizing the risk of signal loss due to channel failure and enhancing the overall reliability and robustness of the system. Besides, comparing a fixed number of channels at different resolutions would require expanding or shrinking the physical coverage area, which introduces confounding factors related to anatomical variability and regional signal heterogeneity. We have revised the relevant sections of the manuscript accordingly, and we are grateful to the reviewer for prompting us to clarify these important points of our methodology.

Figure R1. *t*-SNE visualization of feature distributions extracted from datasets using different axial spatial resolutions. The subsets of 4, 8, 16, and 32 channels are selected varying the axial spatial resolution by adjusting the inter-electrode distance along the longitudinal direction within a fixed electrode array contour.